# Token Alignment Heads: Unveiling Attention's Role in LLM Multilingual Translation

**Binbin Liu**[1], **Wenhan Han**[2], **Feng Chen**[1], **Yifan Zhang**[1], **Ping Guo**[1],
**Haobin Lin**[1], **Bingni Zhang**[1], **Taifeng Wang**[1], **Yin Zheng**[1*]

[1]ByteDance, [2]Eindhoven University of Technology

## Abstract

Recently, large language models (LLMs) have made remarkable progress, with multilingual capability emerging as a core foundational strengths. However, the internal mechanisms by which these models perform translation remain incompletely understood. In this paper, we elucidate the relationship between the attention mechanism in LLMs and their translation abilities. We find that certain attention heads, which we term token alignment heads, are specifically responsible for mapping tokens from the source language to the target language during inference. Through a systematic investigation across various models, we confirm that these token alignment heads exhibit several key characteristics: (1) Universality: They are present in all LLMs we studied. (2) Sparsity: They constitute only a small fraction of all attention heads. (3) Consistency: The set of token alignment heads activated by the model shows strong consistency across different language pairs. (4) Causality: Interventionally removing these heads leads to a sharp decline in the model's translation performance, while randomly removing non-token alignment heads has little impact on translation ability. (5) Functional Specificity: Ablating token alignment heads disproportionately harms translation but has a varied impact on other multilingual tasks. We also traced the formation of token alignment heads during pre-training, revealing an evolutionary path of rapid proliferation, stabilization, and eventual pruning. Furthermore we leverage these token alignment heads to filter multilingual training data, and our experiments show that these data could enhance translation capabilities of the models.

## 1 Introduction

Recently released Large Language Models (LLMs) (Comanici et al., 2025; OpenAI, 2025; Anthropic, 2025; Liu et al., 2024; Yang et al., 2025) have demonstrated remarkable multilingual capabilities, showing significant improvements in both the complexity of multilingual tasks they can handle and the range of languages they support. Multilingual proficiency has now become an essential foundational ability for state-of-the-art LLMs. Among these capabilities, translation is particularly crucial, as it not only represents a key application but also underpins the overall multilingual performance of these models. A deeper understanding of the underlying mechanisms of translation in LLMs is therefore vital, not only for a comprehensive view of their inner workings but also for providing valuable insights to guide the development of superior multilingual training strategies, in terms of both model architecture and data selection.

A growing body of research has begun to demystify how LLMs process multilingual information (Artetxe et al., 2020; Lindsey et al., 2025; Datta et al., 2020; Chang et al., 2022). Several recent works have explored the internal mechanics of multilingualism in LLMs. For instance, Zhao et al. (2024) found that LLMs often initially process queries by converting multilingual inputs into an English-centric representation before solving tasks. Similarly, Schut et al. (2025) revealed that these models tend to make decisions and reason within an English-dominated semantic space. These findings underscore the hypothesis that an internal translation process is a core component of LLM multilingualism, highlighting the importance of understanding these translation mechanisms to further elucidate their broader multilingual capabilities.

---

*Corresponding author. Emails: yzheng3xg@gmail.com

Figure 1: An example of token alignment head pruning in Llama-3.1-8B. The top panel shows the correct translation generated by the original model. The middle panel demonstrates that after the top 30 token alignment heads are pruned, the model loses its ability to translate and reverts to copying the English source text, while its basic copy-paste functionality is preserved. In contrast, the bottom panel shows that pruning 30 random non-token alignment heads (control) has no impact on the translation output.

Prior research (Michel et al., 2019; Vig & Belinkov, 2019; Finlayson et al., 2021; Elhage et al., 2021a) into the attention mechanism has established the functional specialization of individual heads. Early work on Transformer (Vaswani et al., 2023) models for machine translation showed that many attention heads were redundant and could be pruned with minimal impact on performance (Voita et al., 2019; Kovaleva et al., 2019). Subsequent studies, such as Kim et al. (2021); Ma et al. (2021); Zhang et al. (2025), have investigated the role of attention heads in translation by measuring their impact on translation metrics. These studies identified certain attention heads as critical for translation and noted that the sets of important heads are highly similar across different language pairs. However, these approaches often have limitations. They frequently rely on task-specific evaluation metrics, are typically conducted on smaller or single models, and their methods for identifying important heads can be opaque. Crucially, they often stop at identifying which heads are important, without fully explaining how these heads mechanistically contribute to the translation process.

Unlike previous work that measures head importance based on downstream benchmark performance, our approach shifts the focus to identifying the underlying mechanism. Inspired by the discovery of other functionally specialized circuits in LLMs, such as "induction heads" that implement in-context learning (Olsson et al., 2022) and "retrieval heads' designed for knowledge retrieval (Wu et al., 2025a), we hypothesized that a similar specialization must exist for translation. We posited that beyond attention heads performing generic copy-paste behaviors, there must be a set of heads specifically responsible for the core translation task: mapping tokens from a source language to their corresponding tokens in a target language. We term these attention heads, characterized by their direct cross-lingual token alignment behavior which can be regarded as a form of word alignment(Brown et al., 1993; Och & Ney, 2003), "token alignment heads". Figure 1 provides an illustration of this functional specialization: when we ablate the top 30 token alignment heads from a Llama-3.1-8B model (Grattafiori et al., 2024), its translation capability collapses. Crucially, the model does not simply fail; it reverts to a more basic copy-paste behavior, reproducing the English input verbatim. This demonstrates that the model's general ability to copy tokens remains intact, and that the ablated heads perform a specific, non-copying function of cross-lingual mapping.

In this work, we conduct a systematic investigation across a series of LLMs to validate the existence and properties of token alignment heads. We reveal that token alignment heads play a pivotal role in the translation capabilities of LLMs, thereby uncovering the relationship between translation mechanisms and attention. We further identify and validate several key characteristics of these token alignment heads: (1) Universality: All large models we studied possess such token alignment heads; (2) Sparsity: Only a small subset of attention heads function as token alignment heads; (3) Consistency: The token alignment heads activated by the model when translating different language pairs exhibit strong consistency. (4) Causality: Ablating these token alignment heads through causal intervention leads to a significant drop in translation performance, whereas removing a random

equivalent number of non-token alignment heads has little impact. (5) Functional Specificity: Ablating token alignment heads disproportionately harms translation but has a varied impact on other multilingual tasks, suggesting the funcional specificity of token alignment heads. Furthermore, we investigate the formation process of token alignment heads by analyzing the entire pre-training life-cycle of a model. Our analysis reveals a distinct developmental trajectory in three phases: an initial rapid proliferation of token alignment heads that coincides with the acquisition of translation ability, followed by a period where the core set of heads stabilizes, and finally a long phase of consolidation and pruning. This discovery provides insight into how specialized circuits emerge and are optimized during large-scale training.

We further substantiate these findings through a practical application. We introduce TRater, an algorithm that leverages token alignment heads to score multilingual training data based on its importance to the translation mechanism. Our experiments reveal that a small fraction of data identified by TRater is responsible for the model's final translation proficiency. This result provides evidence for the causal role of token alignment heads. The discovery that a core capability like translation is governed by a sparse and functionally specialized circuit provides a concrete target for future research. Ultimately, these insights pave the way for more efficient and robust multilingual systems, enabling targeted architectural innovations, data curation strategies guided by mechanistic understanding.

## 2 DETECTING TOKEN ALIGNMENT HEAD

In this section, we introduce the algorithm for detecting token alignment heads. Since the model involves cross-lingual token alignment during the translation process, we first need to identify the mapping relationship between tokens from the source language to the target language. Then, we define a metric called the translation score to recognize attention heads that implement the model's translation mechanism. The translation score measures the frequency with which an attention head maps tokens from the source language to the target language. If an attention head exhibits a relatively high translation score, it indicates that this attention head frequently performs cross-lingual token alignment when processing different translation texts. Such attention heads are what we refer to as token alignment heads.

### 2.1 TOKEN ALIGNMENT ANNOTATION

Since existing token alignment tools do not cover all languages, we utilize OpenAI's GPT-4.1 model to annotate token alignments in translation texts. Specifically, we require the large language model to identify the corresponding source language token for each target language token and provide a confidence score for each token alignment. To ensure the accuracy of the annotations, we only retain token alignments with a confidence score greater than 0.9. If no corresponding source token exists, it is marked as None. Figure 2 shows an example of token alignment results from English to Chinese.

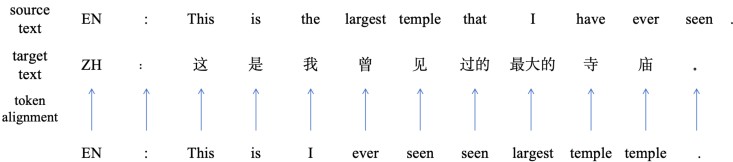

Figure 2: A token alignment example from English text to Chinese text

### 2.2 TRANSLATION SCORE

In the decoding process of translation, we define the translation score as the frequency of valid token alignments by attention heads. Specifically, during the greedy decoding process, let the currently generated token be $t$, and let the attention score of the attention head be denoted as $\mathbf{w} \in \mathbb{R}^{|\mathbf{x}|}$. If token $t$ has a corresponding source token $s$ with position idx as $s_{\mathrm{idx}}$, and the attention head assigns the highest attention score probability to the source token $s$, then we consider that the attention head has successfully completed a language-pair token alignment. Formally, we have: $\mathbf{w}_{s_{\mathrm{idx}}} = \max(\mathbf{w})$.

Let $g_h$ denote the number of valid language-pair token alignments performed by attention head $h$, and let $m$ be the total number of target tokens that have a corresponding valid source token. Then, the translation score of attention head $h$ is defined as:

$$\text{TS}_h = \frac{g_h}{m} \tag{1}$$

## 2.3 TOKEN ALIGNMENT HEAD DETECTION

To empirically identify token alignment heads, we compute the Translation Score for every attention head in the model using the dev split of the FLORES-101 dataset (Goyal et al., 2021). For each of the approximately 900 source-target sentence pairs in a given language direction, we calculate the TS for all heads. The final score for each head is the average TS computed across all examples in that language pair. An attention head is then classified as a token alignment head if its final Translation Score exceeds a predefined threshold of 0.1.

## 3 BASIC PROPERTIES OF TOKEN ALIGNMENT HEADS

In this section, we characterize the fundamental properties of the identified token alignment heads. To ensure the robustness and generalizability of our findings, our analysis spans a diverse set of open-source models. This selection covers a range of parameter sizes (1.7B to 30B), architectures (dense and Mixture-of-Experts (Jacobs et al., 1991)), and training stages (pre-trained and instruction-tuned). The models include Llama-3.1-8B, Mistral-7B-Instruct-v0.3[1], Mistral-7B-v0.3 (Jiang et al., 2023), Qwen2.5-7B (Qwen et al., 2025), Qwen3-1.7B, and Qwen3-30B (Yang et al., 2025).

### 3.1 UNIVERSALITY AND SPARSITY

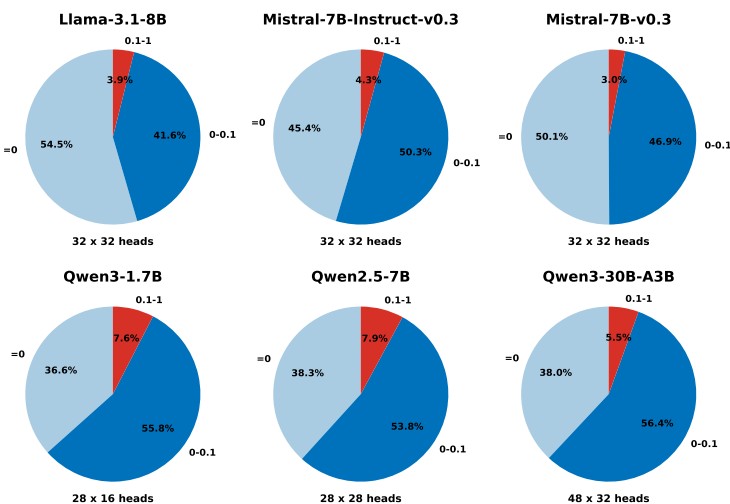

Figure 3: Translation score distribution for different models. According to the different translation scores, the model's attention heads are divided into three categories: token alignment heads (red), infrequently activated heads (blue), and heads with near-zero activation (light blue). All models studied have token alignment heads, and the proportion of token alignment heads is relatively small. Most of the heads are either not activated or are activated at a low frequency.

Our analysis first reveals two fundamental properties: universality and sparsity. As illustrated in Figure 3, token alignment heads (defined as having a translation score $> 0.1$) are present in every model we examined, irrespective of its size, architecture, or training stage. This confirms that they are a universal, emergent feature of multilingual LLMs.

---

[1] https://huggingface.co/mistralai/Mistral-7B-Instruct-v0.3

Concurrently, token alignment heads are exceptionally sparse. They constitute less than $8\%$ of the total attention heads in all models, and as few as $3\%$ in Mistral-7B-v0.3. Additionally, infrequently activated heads during the translation process account for nearly $50\%$. The remaining attention heads, which are almost never activated, account for between $36\%$ and $55\%$.

We next analyze the positional distribution of these heads within the model architecture. Figure 4 shows a consistent pattern: token alignment heads are predominantly concentrated in the middle layers of the models. In contrast, the earliest and latest layers contain very few token alignment heads. This aligns with the broader understanding of Transformer architectures, where initial layers are thought to handle surface-level feature extraction and final layers are responsible for structuring the output.

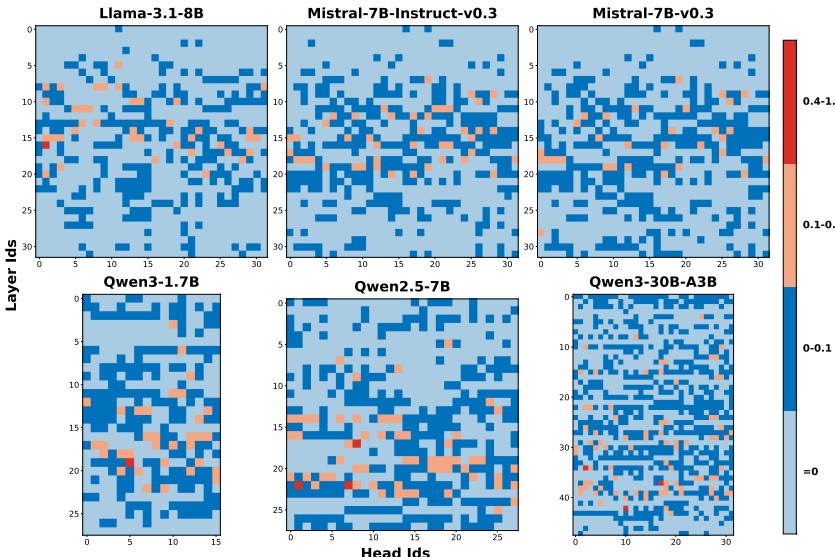

Figure 4: Positional distribution of translation scores in different models. Each heatmap visualizes the translation score distribution. The color intensity corresponds to the translation score, with warmer colors (red/orange) indicating higher scores and cooler colors (blue) indicating lower scores.

## 3.2 LANGUAGE CONSISTENCY

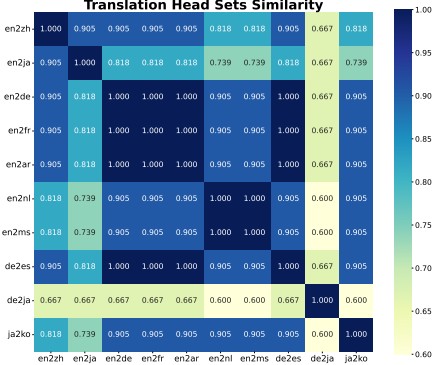

Figure 5: Jaccard similarity matrix of token alignment head sets across various language pairs. The axes list ten different language pairs. The token alignment heads between different language-pairs are very similar, most of the similarity scores are above 0.9.

To investigate the consistency of token alignment heads in the model across different language pairs, we selected the following language pairs: English-Chinese (en2zh), English-Japanese (en2ja), English-German (en2de), English-French (en2fr), English-Arabic (en2ar), English-Dutch (en2nl), English-Malay (en2ms), German-Spanish (de2es), German-Japanese (de2ja), and Japanese-Korean (ja2ko). These pairs cover a variety of linguistic families. Without loss of generality, we focus this analysis on the Llama-3.1-8B model. For each language pair, we selected the top 20 token alignment heads to form its translation set. Then we compute the pairwise similarity between these sets using the Jaccard index:

$$\text{Sim}_{S,T} = \frac{|S \cap T|}{|S \cup T|} \tag{2}$$

As shown in Figure 5, the similarities between all language pairs are relatively high, with Jaccard similarity scores consistently exceeding 0.8 for most pairs, and never dropping below 0.6. This indicates that a largely invariant set of attention heads is responsible for translation across diverse linguistic families, demonstrating the strong cross-lingual generalizability of token alignment heads.

## 4 FORMATION PROCESS OF TOKEN ALIGNMENT HEADS

This section investigates the formation process of token alignment heads during the model's pre-training lifecycle. To trace their development, we trained an 8B parameter model, architecturally identical to Llama-2 model (Touvron et al., 2023), from scratch. The model was trained for a total of 15 trillion tokens. The specific composition of the dataset and the hyperparameters for training are detailed in Appendix A.1. To understand the formation process of token alignment heads, we analyzed model checkpoints at multiple intervals throughout training to map the evolutionary trajectory of token alignment heads.

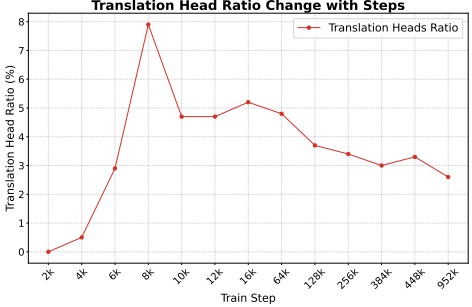
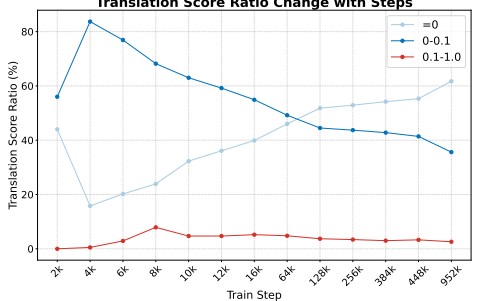

(a) The ratio of token alignment heads rises sharply in the early training stages, reaching a peak of approximately 8% at 8k step. Following this peak, the proportion drops and stabilizes around 5% between 10k and 64k steps. Subsequently, it enters a long phase of gradual decline, settling at 2.6% by the end of training.

(b) The distribution of all attention heads by activity level. The proportion of inactive heads (TS = 0, light blue line) starts at a low point but steadily increases throughout training, reaching over 60% by the final step. Conversely, the proportion of low-activity heads (dark blue line) begins as the dominant group but consistently decreases over time.

Figure 6: The evolutionary trajectory of token alignment heads during training

**Phase 1: Rapid Proliferation** (Early Training Stage, 0-8k steps). In the initial stages of training, as shown in Figure 6(a), the proportion of token alignment heads experiences a rapid proliferation, growing from zero to its peak. This period of rapid circuit formation coincides directly with the steepest gains in the model's translation performance, where the FLORES chrF++ metric surged from 12.58 to 45.77. This suggests that the initial acquisition of translation ability is contingent on the rapid emergence of these specialized heads.

**Phase 2: Set Stabilization** (Early-to-Mid Training Stage, 10k-64k steps). From 10k to 64k steps, the proportion of token alignment heads stabilized around 5%, and the core set of these heads becomes remarkably stable. To quantify this, we define a conditional overlap ratio metrics which measures the overlap between the token alignment head set at any given step (A) and the final refer-

ence set at the end of training (B):

$$\frac{|A \cap B|}{|B|} \qquad (3)$$

As shown in Figure 7, from approximately 8k steps onward, conditional overlap ratio remains consistently high. This indicates that the token alignment heads formed rapidly in the early stage of training are largely maintained throughout the subsequent training process.

**Phase 3: Consolidation and Pruning** (Mid-to-Late Training Stage, 64k-952k steps). In this longest phase of training, we observe a gradual decline in the overall proportion of token alignment heads, which settles at 2.6% (Figure 6(a)). Given that the core set of heads remains stable (Phase 2), this decline implies that heads with weaker or more redundant translation capabilities are being "pruned"—their translation scores fall below the threshold as the network refines its functions.

We hypothesize this pruning is part of a broader network-wide optimization towards increased sparsity and computational efficiency. This is corroborated by the shifting distribution of head activity shown in Figure 6(b). As training progresses, the proportion of completely inactive heads steadily increases, reaching 61.7% by the end. This happens at the expense of low-to-moderately active token alignment heads. In essence, the model learns to solve the translation task not by using more heads, but by relying more heavily on a smaller, more efficient, and highly specialized set of circuits, while deactivating others. This process of over-producing and then refining specialized circuits appears to be a key mechanism in the development of efficient neural networks.

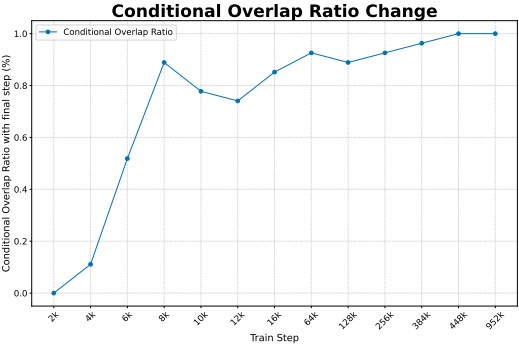

Figure 7: Stability of the token alignment head set over time. The conditional overlap ratio exhibits a steep and rapid increase during the initial training phase, rising from zero to nearly 0.9 at 8k step. From this point onward, the overlap remains consistently high, fluctuating but generally staying above 0.8 and approaching 1.0 by the end of training.

## 5    INFLUENCE ON DOWNSTREAM TASKS

In this section, we investigate the impact of token alignment heads on downstream benchmarks. First, we analyze the influence of token alignment heads on the model's translation performance to demonstrate their causality. Here, we select the FLORES101 benchmark to evaluate the model's translation performance. Next, we examine the impact of token alignment heads on the model's general multilingual capabilities. We evaluate the following benchmarks: translated Hellaswag (Zellers et al., 2019), ARC-Easy and ARC-Challenge (Clark et al., 2018) which are detailed in Appendix A.2, XWinograd (Tikhonov & Ryabinin, 2021), XStoryCloze (Mostafazadeh et al., 2016), XNLI (Conneau et al., 2018), XCOPA (Ponti et al., 2020), and a localized multilingual variant of the MMMLU[2] test set denoted as XMMLU which includes JMMLU[3], CMMLU (Li et al., 2024), AMMLU[4], IndoMMLU (Koto et al., 2023), and VMLU[5]. We study the influence of token alignment heads by mask those token alignment heads.

---

[2]https://huggingface.co/datasets/openai/MMMLU

[3]https://huggingface.co/datasets/nlp-waseda/JMMLU

[4]https://huggingface.co/datasets/Hennara/ammlu

[5]https://vmlu.ai

## 5.1 TRANSLATION CAPACITY

In this subsection, we investigate the impact of token alignment heads on the translation performance of Llama-3.1-8B, Mistral-7B-v0.3, Qwen2.5-7B, Qwen3-1.7B, and Qwen3-30B using the FLORES-101 benchmark. We compare translation metrics before and after masking token alignment heads, as well as after masking random non-token alignment heads. To clearly illustrate the effects, we report the difference between the metrics of the masked models and those of the baseline models (without masking). Figure 12 presents the changes in BLEU and chrF++ scores. Masking token alignment heads leads to substantial declines in both metrics, with the largest drops exceeding 17 points for BLEU and 25 points for chrF++. In contrast, masking random heads has only a minimal effect. These results demonstrate that token alignment heads have a direct and significant influence on the models' translation capabilities, a property we refer to as the causality of token alignment heads.

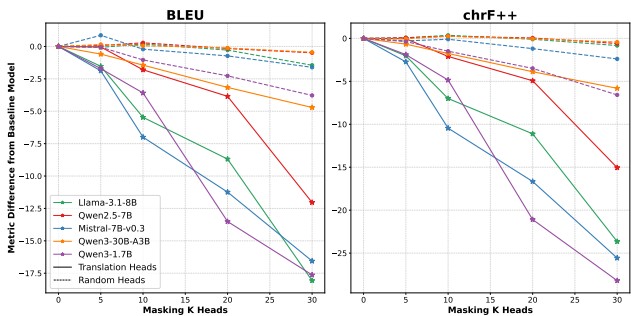

Figure 8: Impact of masking token alignment heads versus random heads on FLORES benchmark scores. Masking token alignment heads leads to a significantly larger performance drop compared to masking random heads

## 5.2 MULTILINGUAL CAPACITY

We then investigate the impact of ablating token alignment heads on a broader suite of multilingual benchmarks. From the results presented in Figure 9, we observe a clear hierarchy of dependency on token alignment heads. Benchmarks such as Hellaswag_ML, ARC_C_ML, and ARC_E_ML exhibit a significant performance drop (up to 10 points), which is consistently larger than the drop from ablating random heads. This suggests that these tasks, while not pure translation, partially rely on the cross-lingual mapping capabilities provided by token alignment heads. This functional overlap may stem from translation artifacts in their data creation process or a genuine need for cross-lingual conceptual alignment to solve the tasks.

In contrast, for other benchmarks like XNLI and XCOPA, token alignment heads demonstrate weak causality, as their ablation often results in a smaller performance drop than that of random ablation. This indicates that these tasks depend on different multilingual mechanisms within the model, likely operating at a higher semantic level that does not require the token-level mapping performed by token alignment heads. These findings suggest that token alignment heads provide a foundational cross-lingual alignment capability that various downstream tasks leverage to different degrees.

## 5.3 TOKEN ALIGNMENT HEAD AS DATA RATER

To further probe the relationship between token alignment heads and multilingual data, we introduce TRater, a data-filtering algorithm. TRater leverages token alignment heads to score data samples based on their importance to the translation mechanism. We compute the score of sample $x$ as follows:

$$\text{score}(x) = \frac{1}{m} \sum_i \left( L(\theta_{\text{mask}}, x_i) - L(\theta, x_i) \right) \tag{4}$$

where $L$ denotes the token level cross entropy loss, $\theta$ denotes the original model parameters, $\theta_{\text{mask}}$ represents the model parameters after masking the top 20 token alignment heads, $i$ is the token

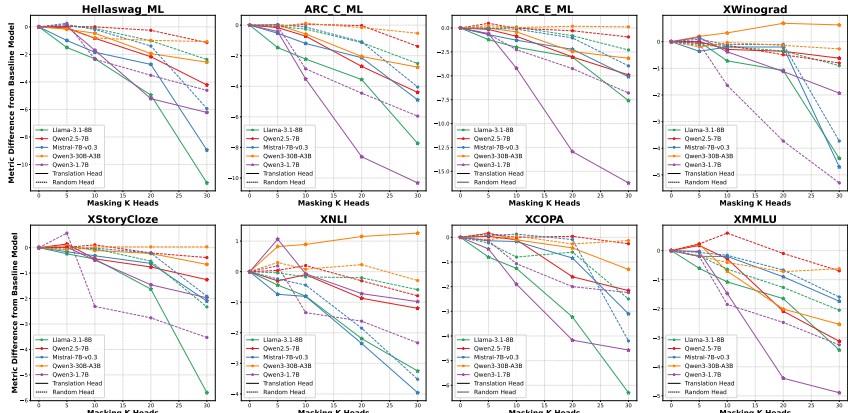

Figure 9: Performance (Accuracy) change across multilingual benchmarks when masking token alignment heads versus random heads. Tasks like Hellaswag_ML and ARC_ML show substantial drops when token alignment heads are ablated, while others such as XNLI and XCOPA are less affected, reflecting varying degrees of reliance on cross-lingual alignment.

index in $x$, and $m$ is the total number of tokens. This score quantifies the performance degradation on a sample when token alignment heads are removed, with higher scores indicating greater reliance on these heads. We conduct experiments on the 1.5B model, training on a total of 1T tokens. The dataset comprises 700B tokens of English web data and 300B tokens of multilingual web data. Using the TRater algorithm, we score the 300B multilingual data, and we select the top $1.3\%$ for each language. We design the following two experiments to validate the impact of the selected data:

**Remove**: From the baseline multilingual datas, we exclude the selected data. To maintain the data proportions unchanged, we increase the volume of the remaining data per language. And we ensure no additional duplicates compared to the baseline.

**Enhance**: The selected data is triplicated, while the remaining data is proportionally down-sampled to preserve the overall language distribution.

Table 1: Performance of baseline, remove, and enhance settings across multiple metrics.

| Model | flores_chrF++ | MMMLU | Hellaswag_ML | ARC_C_ML | ARC_E_ML | XWinograd | XStoryCloze | XNLI | XCOPA | XMMLU |
|---|---|---|---|---|---|---|---|---|---|---|
| baseline | 43.87 | **26.89** | 44.88 | 31.40 | 53.97 | **75.70** | 58.40 | 41.94 | 62.88 | 30.71 |
| remove | 41.33 | 26.58 | 44.69 | 31.37 | 54.54 | 73.63 | 58.15 | **42.10** | 63.20 | 30.84 |
| enhance | **46.68** | 26.71 | **44.95** | **31.54** | 54.94 | 74.33 | **58.44** | 41.72 | **63.70** | **30.88** |

The experimental results are presented in Table 1. From the table, we observe that the data filtered by token alignment heads is crucial for the model's translation capabilities: the **Remove** setup exhibits a noticeable decline in translation performance compared to the baseline, whereas the **Enhance** setup shows a observable improvement.

However, the impact of this selected data on other multilingual benchmarks is less pronounced than its effect on FLORES. This can be attributed to two main factors. Firstly, the performance change observed in FLORES due to the selected data (about 2-3 points) is substantially smaller than the drops seen when ablating token alignment heads (over 10 points). Consequently, any impact on other multilingual benchmarks becomes less perceptible. Secondly, while the function of token alignment heads is partially leveraged by some non-translation tasks, these benchmarks only require the model to possess a foundational translation capability. Once this sufficient baseline is established, further increasing the proportion of translation-centric data yields diminishing returns for the model's general multilingual performance. Indeed, a qualitative analysis confirms the selected data is highly translation-specific, consisting predominantly of bilingual corpora. Several representative examples are detailed in Appendix A.3.

## 6 Related Work

A substantial body of work exists on understanding the internal mechanisms of large language models. Based on the granularity of analysis, we broadly categorize these studies into three main areas: semantic space, neuron-level mechanisms, and head-level mechanisms.

### 6.1 Semantic Space

Prior works (Wendler et al., 2024; Schut et al., 2025; Zhao et al., 2024; Wu et al., 2025b; Harrasse et al., 2025) study the geometry of multilingual representations and often concludes that models "think in English" or in a shared latent semantic space in middle layers. These works explain where multilingual information lives and how information exists (English or Language-Agnostic Space). Our results are complementary: we identify token alignment heads concentrated in similar middle layers and show that they implement token-level cross-lingual alignment, routing the aligned source token's representation into the target position.

### 6.2 Neuron-level mechanisms

Utilizing causal mediation analysis across a diverse range of in-context learning (ICL) tasks, Todd et al. (2024) identified a key mechanism termed "function vectors", which trigger the model to execute specific procedural tasks. Similarly, Wang et al. (2024) employed causal mediation procedures to locate attention heads pivotal for machine translation, leveraging these heads to construct translation vectors that mitigate language mismatch errors. In contrast, our work shifts focus from the task-triggering level to the token-execution level. We find that token alignment heads facilitate the actual cross-lingual alignment. Analogously, if the function vector acts as the "master switch" activating the translation mode, the token alignment heads are the vital machinery carrying out the translation itself.

From the perspective of languages, some studies (Liu et al., 2025; Zhao et al., 2024) pinpoint neurons that are specialized for encoding language identity and language-specific features and shows that ablating or fine-tuning them selectively affects particular languages. This explains which subcircuits are responsible for "being in language X". In contrast, we operate at the head level and focus on the cross-lingual alignment step, i.e., how information moves between languages during translation.

### 6.3 Head-level mechanisms

Works on induction heads, retrieval heads, and circuits (Elhage et al., 2021b; Olsson et al., 2022; Wu et al., 2025a; Bricken et al., 2024; Zhang et al., 2024) shows that a small number of specialized heads can explain non-trivial capabilities such as in-context learning or long-context retrieval. Recent work (Liu et al., 2025; Zhang et al., 2025) identifies language heads or translation-related heads by ranking heads via their impact on downstream loss, perplexity, or logits on specific benchmarks, sometimes using path patching. Our approach is closely related but uses a different identification signal: we define Token Alignement Heads (TAH) using alignment-based translation score—heads are selected because they consistently link target tokens to their externally aligned source tokens, independent of any particular evaluation task. Masking experiments are then used only as a causal validation step. This makes our notion of specialization explicitly lexical and cross-lingual.

## 7 Conclusion

In this paper, we identified a special class of attention heads responsible for mapping source language tokens to target language tokens during translation. We experimentally confirmed that these heads are universal, consistent, and have a direct causal effect on the model's translation capabilities. We also uncovered their evolutionary process during pre-training, which involves rapid formation, stabilization, and pruning. More importantly, We found that a tiny fraction of critical data filterd by token alignment heads, proves decisive for translation performance but its impact on other multilingual tasks is less pronounced. This finding suggests that translation operates as a separable module within LLMs. Our work pave the way for more efficient and robust multilingual systems, enabling targeted architectural innovations, data curation strategies guided by mechanistic understanding.

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

# A APPENDIX

## A.1 TRAINING SETTING

In Section 4, We use AdamW (Loshchilov & Hutter, 2019) optimizer to train a 8B dense model with a structure identical to Llama-2 model Touvron et al. (2023). The training dataset, totaling 15 trillion tokens, comprises English and multilingual data sourced from cleaned and filtered ccwarc, along with open-source resources including Wikipedia, books, academic papers, mathematics, code, and parallel corpora. The hyperparameters were set as follows: learning rate (lr) = $3.6 \times 10^{-4}$, global batch size (gbs) = 4096, sequence length = 4096, weight decay = 0.1. We employed a cosine learning rate scheduler that decayed to 0.1 of the peak learning rate. For our analysis, we selected model checkpoints at 2,000, 4,000, 6,000, 8,000, 12,000, 16,000, 64,000, 128,000, 256,000, 384,000, 448,000, and 952,000 steps (near the end of training).

In Section 5.3, we trained the 1.5B model using the AdamW optimizer with the following hyperparameters: global batch size (gbs) = 4096, sequence length = 4096, weight decay = 0.1, learning rate (lr) = $5.0 \times 10^{-4}$, and with lr cosine decay to $5.0 \times 10^{-5}$, the multilingual web data includes a total of 17 languages, specifically including German, Spanish, French, Indonesian, Thai, Korean, Vietnamese, Arabic, Turkish, Italian, Malay, Chinese, Portuguese, Japanese, Dutch, Russian and Filipino. The proportion of different languages is determined by the method in Guo et al. (2025).

## A.2 TRANSLATED BENCHMARK

For evaluating Hellaswag_ML, ARC_C_ML and ARC_E_ML, we use the MuBench dataset (Han et al., 2025). The evaluation covers 18 languages represented in our training data, namely: English, German, Spanish, French, Indonesian, Thai, Korean, Vietnamese, Arabic, Turkish, Italian, Malay, Chinese, Portuguese, Japanese, Dutch, Russian and Filipino.

## A.3 TEXT CASES FOR TRATER

Table 2: Text cases filtered by TRater for German, French and Spanish.

| Language | Example |
|---|---|
| **German** | Zwischen zwei Seen, die unterschiedlicher kaum sein können, liegt Wandlitz. Der Wandlitzsee, bebaut, kaum zugänglich mit unzähligen Wassergrundstücken, der Liebnitzsee, frei zugänglich, Badestelle, Fähre zur Insel und Naherholungsgebiet im Buchenwald... Wandlitz is located between two lakes that could hardly be more different. The Wandlitzsee, built-up, hardly accessible with countless water properties, the Liebnitzsee, freely accessible, bathing area, ferry to the island and local recreation area in the beech forest... |
| **French** | Bruno Houssin, designer français et professeur à l'école de design de Nantes. Diplômé de l'école Boulle de Paris, en Architecture intérieure et Design en 1986... Bruno Houssin, French designer and teacher at the Nantes School of Design. Graduated from the Boulle school of Paris, in Interior Architecture and Design in 1986... |
| **Spanish** | Las placas tectónicas son como grandes balsas que se reparten por toda la corteza del planeta. Unas son de carácter continental, otras de carácter oceánico, contando las primeras con un espesor mayor que el de las segundas... Le tremblement de terre en Haïti est partie de l'ensemble de la libération des tensions accumulées à l'occasion du mouvement des plaques tectoniques dans les Caraïbes et en Amérique du Nord... |

Table 3: Text cases filtered by TRater for Italian, Portuguese, Chinese and Dutch.

| Language | Example |
|---|---|
| **Italian** | Dorothy Bhawl è un artista autodidatta interessato al mondo contemporaneo, soprattutto alla realtà che appartiene e avvolge questa epoca: quello della comunicazione, social network, spiritualità e grottesco con un sentimento di odi et amo... Dorothy Bhawl is a self-taught artist interested in the contemporary world, especially in the reality that belongs and envelops this era: that of communication, social networks, spirituality and grotesque with a feeling of hatred and love... |
| **Portuguese** | André Rigatti Centro Universitário Maria Antonia USP Sempre próximas a suas bordas, as pinturas de André Rigatti possuem pequenas aberturas, por onde se deixa ver o processo que dá origem a trabalhos de textura matérica mais ou menos acentuada, resultados da sobreposição de diversas camadas de tinta, aplicadas cada uma seguindo uma direção diferente do pincel... Always close to its edges, André Rigatti's paintings have small openings, where you can see the process that gives rise to more or less accentuated texture work, results of the overlapping of several layers of paint, applied each following a direction... |
| **Chinese** | 原文: 版印书籍，唐人尚未盛为之。自冯瀛王始印五经，已后典记，皆为版本。庆历中，有布衣毕升，又为活版。其法用胶泥刻字，薄如钱唇，每字为一印，火烧令坚。先设一铁板，其上以松脂腊和纸灰之类冒之。欲印则以一铁范置铁板上，乃密布字印。满铁范为一板，持就火炀之，药稍熔，则以一平板按其面，则字平如砥... 译文：用刻板印刷书籍，唐朝人还没有大规模采用它。五代始才开始印刷五经，以后的各种图书都是雕板印刷本。庆历年间，有位平民毕升，又创造了活板。他的方法是用胶泥刻成字，字薄得像铜钱的边缘，每个字制成一个字模，用火来烧使它坚硬。先设置一块铁板，它的上面用松纸、蜡混合纸灰这一灰东西覆盖它... |
| **Dutch** | Kees Blom (Apeldoorn, 1968) komt uit een artistieke familie. Zijn vader had al een passie voor de schilderkunst maar pas zoon Kees lukt het om de stap naar zelfstandig kunstschilder te zetten...Kees Blom (Apeldoorn, 1968) comes from an artistic family. While his father already had a passion for painting, son Kees succeeds in taking the step to become an independent painter... |

## A.4 CASE STUDY FOR TOKEN ALIGNMENT HEADS PRUNING

To clarify the role of Token Alignment Heads (TAHs), we conducted a systematic and detailed analysis of the model's performance on translation tasks when TAHs are masked (specifically focusing on cases where the model translated correctly before masking TAHs but failed to do so afterwards). We observed that the failure modes resulting from masking THs can be broadly categorized into three types:

1. No Translation (46%) The model fails to generate the target language and merely repeats the source text content. As illustrated in Figure 10, masking the Token Alignment Heads completely disables the model's word alignment capability for certain queries, preventing the generation of target language output.

2. Missing Details (36%) The translated output lacks specific details found in the source text. In these instances, the model's word alignment capability is partially compromised. While the model can still generate text in the target language, it fails to correctly map certain source information to the target output. This results in the omission of key details, leading to incomplete or imprecise

---

Translate this sentence from English to Simplified Chinese
Sentence: In late 2017, Siminoff appeared on shopping television channel QVC.
**[No Heads Pruned]** Answer: 2017年末，辛明诺夫出现在购物电视频道QVC上。
**[30 Token Alignment Heads Pruned]** Answer: In late 2017, Siminoff appeared on shopping television channel QVC.
**[30 Random Heads Pruned]** Answer: 在2017年晚些时候，Siminoff出现在购物电视频道QVC上。

Sentence: One antibody cocktail, ZMapp, initially showed promise in the field, but formal studies indicated it had less benefit than sought in preventing death.
**[No Heads Pruned]** Answer: 一种抗体混合物，ZMapp，最初在野外显示出有希望，但正式研究表明它比预期的更少有益于预防死亡。
**[30 Token Alignment Heads Pruned]** Answer: One antibody cocktail, ZMapp, showed promise in the field, but formal studies indicated it had less benefit than sought in preventing death.
**[30 Random Heads Pruned]** Answer: 一种抗体混合物，ZMapp，最初在野外表现出希望，但正式研究表明，它比预期预防死亡的作用要小。

Sentence: Scientists say this animal's plumage was chestnut-brown on top with a pale or carotenoid-colored underside.
**[No Heads Pruned]** Answer: 科学家说，这种动物的羽毛上面是棕色，下面是淡黄色或是橙色。
**[30 Token Alignment Heads Pruned]** Answer: Scientists say this animal's plumage was chestnut-brown on top with a pale or carotenoid-colored underside.
**[30 Random Heads Pruned]** Answer: 科学家们说，这种动物的羽毛是棕色顶部和淡色或橘色底部。

Figure 10: No Translation cases for masking token alignment heads. For the queries shown in the figure, after masking the token alignment heads, the model no longer outputs content related to the target language, but simply repeats the source text.

translations. Figure 11 presents three such cases, where the text highlighted in green represents the details omitted by the model.

3. Hallucination (18%) The translated output contains content that is completely absent from the source text. This is likely because, without token alignment capabilities, the model cannot directly associate source tokens with target tokens. Consequently, it resorts to generating content in the target language space that is only loosely or tangentially related to the source tokens. Figure 12 displays three examples of this category, where the text highlighted in red indicates output generated by the model that is entirely unrelated to the source text.

---

Translate this sentence from English to Simplified Chinese
Sentence: Like some other experts, he is skeptical about whether diabetes can be cured, noting that these findings have no relevance to people who already have Type 1 diabetes.
**[No Heads Pruned]** Answer: 他和其他专家一样，对糖尿病能否治愈持怀疑态度，指出这些发现对已经患有1型糖尿病的人没有任何意义。
**[30 Token Alignment Heads Pruned]** Answer: 他认为这些发现对那些已经有了糖尿病的人没有什么帮助。
**[30 Random Heads Pruned]** Answer: 和其他专家一样，他对糖尿病能否治愈持怀疑态度，指出这些发现对已经患有1型糖尿病的人没有任何意义。

Sentence: This not only confirms that at least some dinosaurs had feathers, a theory already widespread, but provides details fossils generally cannot, such as color and three-dimensional arrangement.
**[No Heads Pruned]** Answer: 这不仅证实了至少有一些恐龙有羽毛，一个已经广泛流传的理论，而且提供了化石通常不能提供的细节，如颜色和三维排列。
**[30 Token Alignment Heads Pruned]** Answer: 这不仅证实了至少有一种恐龙有羽毛，而且还证实了这种羽毛的颜色和羽毛的三维结构。
**[30 Random Heads Pruned]** Answer: 这不仅证实了至少有些恐龙有羽毛——这一理论已经广为流传——而且还提供了化石通常无法提供的细节，比如颜色和三维排列。

Sentence: During his trip, Iwasaki ran into trouble on many occasions.
**[No Heads Pruned]** Answer: 在他旅行期间，伊瓦斯基遇到了很多麻烦。
**[30 Token Alignment Heads Pruned]** Answer: 在他旅行的过程中，他遇到了很多麻烦。
**[30 Random Heads Pruned]** Answer: 在旅行期间，伊瓦斯基遇到了很多麻烦。

Figure 11: Cases of Missing Details for masking token alignment heads. For the queries shown in the figure, after masking the token alignment heads, the model is still able to generate target language content. However, because some cross-lingual token alignment capabilities are missing, the model loses certain details present in the source text.

Translate this sentence from English to Simplified Chinese
Sentence: During the 1976 selections he advised Carter on foreign policy, then served as National Security Advisor (NSA) from 1977 to 1981, succeeding Henry Kissinger.
[No Heads Pruned] Answer: 1976年选举期间，他为卡特提供建议，1977年至1981年担任国家安全顾问（NSA），接替亨利·基辛格。
[30 Token Alignment Heads Pruned] Answer: 1976年，卡特担任美国总统期间，布什担任国务卿，为卡特提供建议。
[30 Random Heads Pruned] Answer: 1976年选举期间，他为卡特总统提供了外交政策建议，1977年至1981年期间，他担任国家安全顾问（NSA），接替亨利·基辛格。

Sentence: The vehicle itself was taken away from the scene of the accident at approximately 1200 GMT on the same day.
[No Heads Pruned] Answer: 车辆本身在事故现场被拖走，时间大约是当天下午12点。
[30 Token Alignment Heads Pruned] Answer: 该车辆在事故发生后被送往了1200公里以外的地方。
[30 Random Heads Pruned] Answer: 车辆本身在事故现场被拖走，时间大约是当天下午12点。

Sentence: Several large television screens were installed in various places in Rome to let the people watch the ceremony.
[No Heads Pruned] Answer: 罗马市内各处安装了几大块电视屏幕，让人们观看仪式。
[30 Token Alignment Heads Pruned] Answer: 在罗马，人们可以在许多地方看到电视节目。
[30 Random Heads Pruned] Answer: 罗马各地安装了数个大型电视屏幕，供人们观看仪式。

Figure 12: Hallucination cases for masking token alignment heads. For the queries shown in the figure, after masking the token alignment heads, the model exhibits hallucinations, specifically generating output that contains information not present in the source text.

## A.5 BASIC PROPERTIES OF TOKEN ALIGNMENT HEADS ACROSS COMPREHENSIVE MODEL FAMILIES

To provide a more thorough and comprehensive evaluation of Token Alignment Heads, we conducted experiments across four well-established multilingual LLM families: Llama3, Qwen3, Mistral, and Gemma2. Our experimental design ensures broad coverage by including models of varying scales within each family: small (1B/2B parameters), medium (8B/9B parameters), and large (over 13B parameters). Where applicable, we evaluate the instruction-tuned (Instruct) variants. Additionally, for families that feature Mixture-of-Experts (MoE) architectures, our study includes a specific analysis of these models.

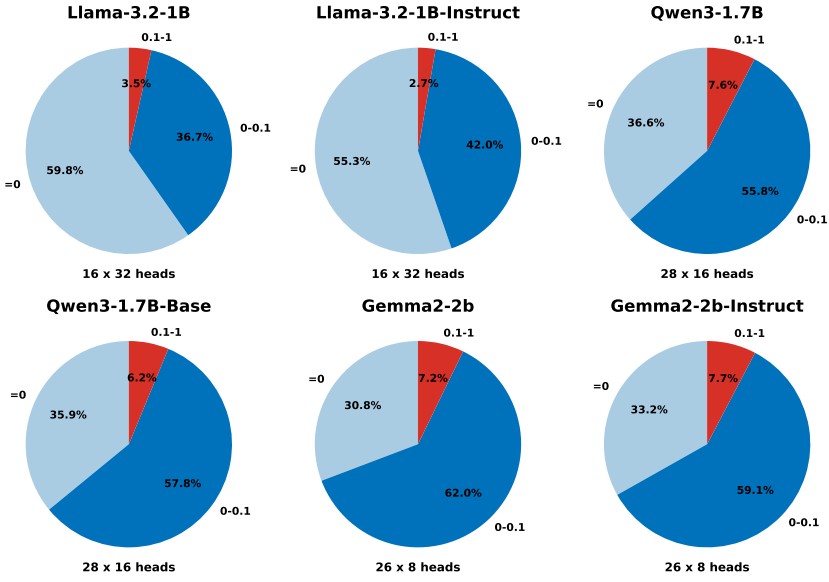

Figure 13: Translation score distribution for small size group models.

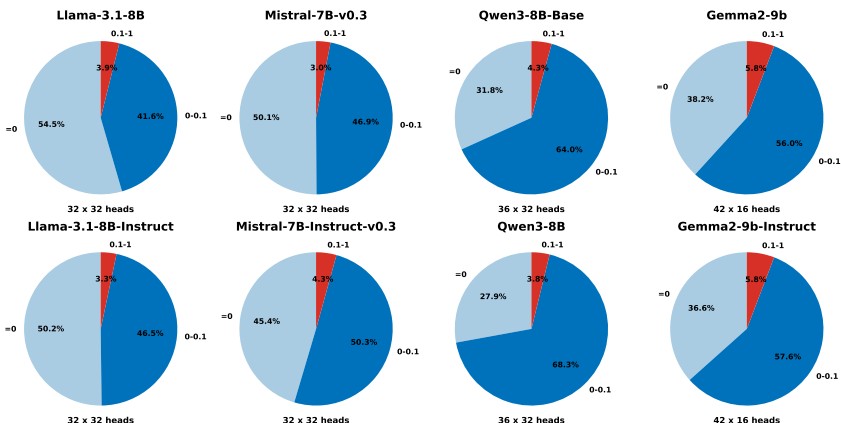

Figure 14: Translation score distribution for medium size group models.

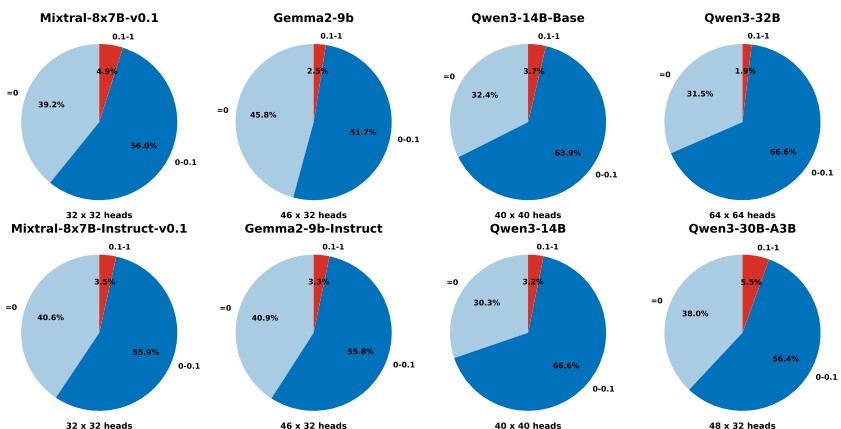

Figure 15: Translation score distribution for large size group models.

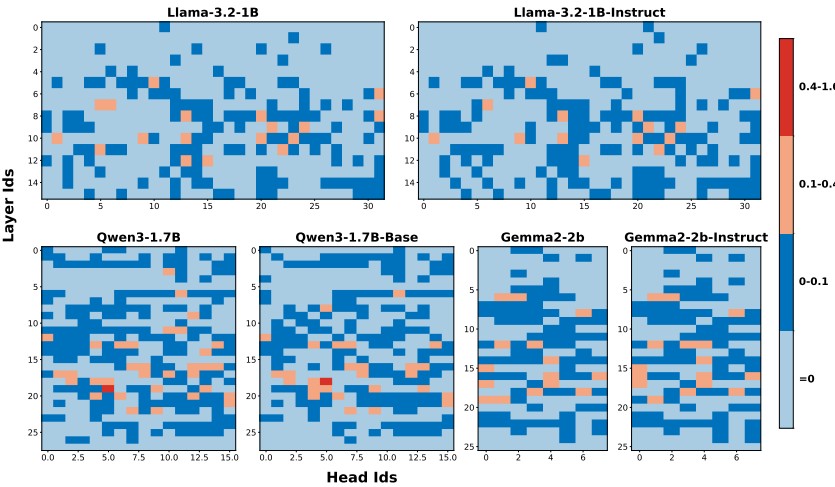

Figure 16: Positional distribution of translation scores in small size group models.

For clarity in presenting our experimental findings, we have organized the models into three distinct groups according to their scale: small, medium, and large. (The Mixtral-8x7B model is classified

within the medium-scale group, as its quantity of attention heads is analogous to that of other models in this tier.) Our initial analysis focuses on demonstrating the basic characteristics of Token Alignment Heads. As illustrated in Figure 13, Figure 14 and Figure 15, Token Alignment Heads are a pervasive phenomenon, consistently identified across all models under investigation—irrespective of model scale, architecture (dense versus MoE), or training paradigm (Base versus Instruct). Crucially, these heads universally demonstrate the property of sparsity. Additionally, Figure 16. Figure 17 and Figure 18 illustrates the positional distribution of token alignment heads across these models. The general distribution pattern observed aligns with the description provided in the main body of the paper.

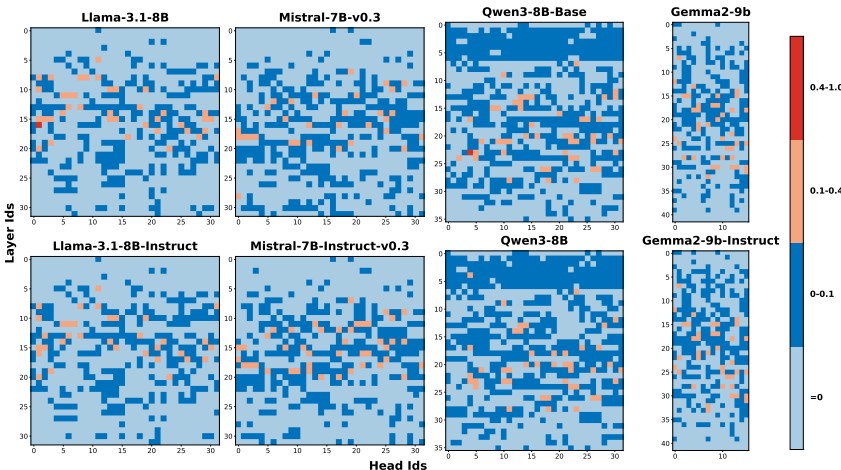

Figure 17: Positional distribution of translation scores in medium size group models.

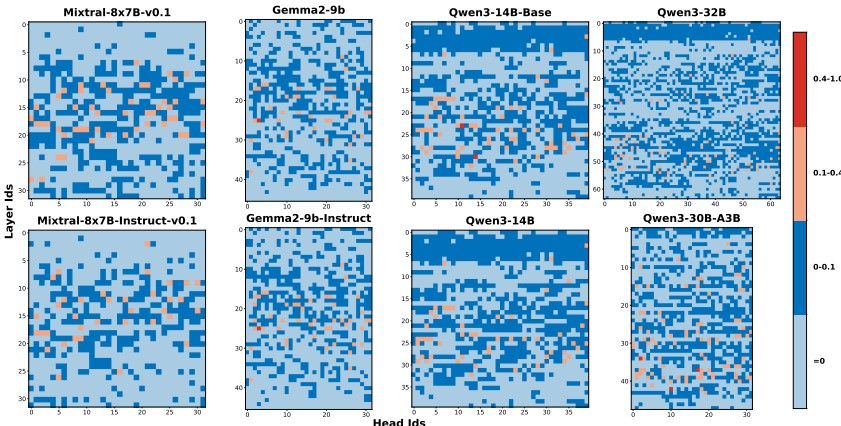

Figure 18: Positional distribution of translation scores in large size group models.

## A.6 DOWNSTREAM TASKS INFLUENCE ACROSS COMPREHENSIVE MODEL FAMILIES

In this section, we present the impact of token alignment heads on downstream task performance, analyzed on a group-by-group basis. To better evaluate translation performance, we supplement the BLEU and chrF++ metrics from the main text with two additional metrics: BLEURT and COMET.

As can be seen, the experimental conclusions are consistent with those in the main text. As illustrated in Figure 19, Figure 20 and Figure 21, for BLEURT and COMET, masking token alignment heads leads to a significant drop in scores, whereas masking random heads results in only a minimal decrease. This demonstrates the causal role of token alignment heads in translation capability. For Hellaswag_ML, ARC_C_ML, and ARC_E_ML, masking token alignment heads causes a larger performance drop than masking random heads, but the overall magnitude of the decrease is far less pronounced than that for the translation metrics. This suggests that these metrics rely to some extent on token alignment capabilities. In contrast, for metrics like XWinograd and XNLI, performance after masking token alignment heads can be better than after masking random heads, indicating that these metrics prioritize other model abilities, such as reasoning, over translation capability.

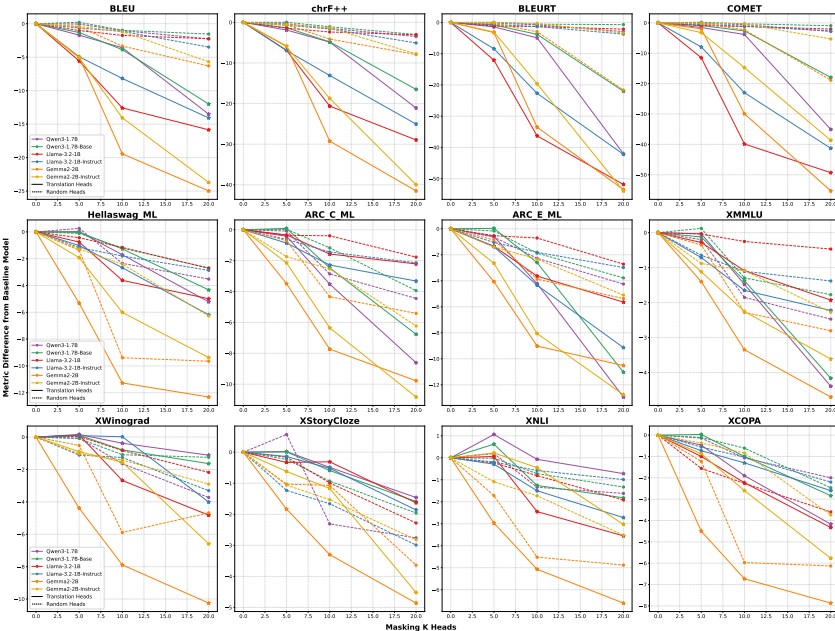

Figure 19: Performance change across downstream benchmarks for small size group models when masking token alignment heads versus random heads.

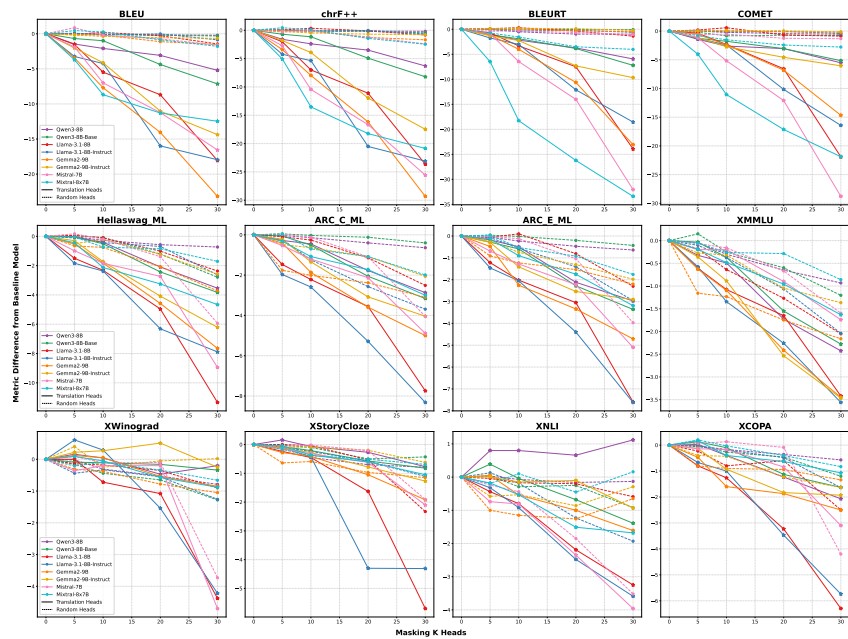

Figure 20: Performance change across downstream benchmarks for medium size group models when masking token alignment heads versus random heads.

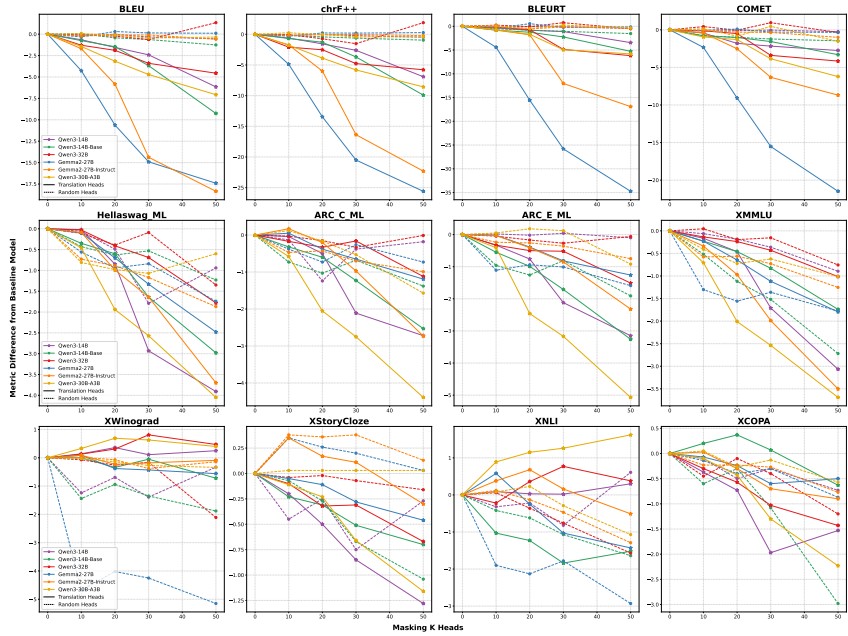

Figure 21: Performance change across downstream benchmarks for large size group models when masking token alignment heads versus random heads.

## A.7 THE ROLE OF TOKEN ALIGNMENT HEADS ON MULTILINGUAL TASKS

In this section, we showcase the utility of Token Alignment Heads on multilingual tasks through a series of case studies. To elucidate their performance characteristics under varying conditions, we analyze two distinct scenarios: the Hellaswag_ML task, which is moderately dependent on Token Alignment Heads, and the XNLI/XWinograd tasks, where the dependency is substantially weaker.

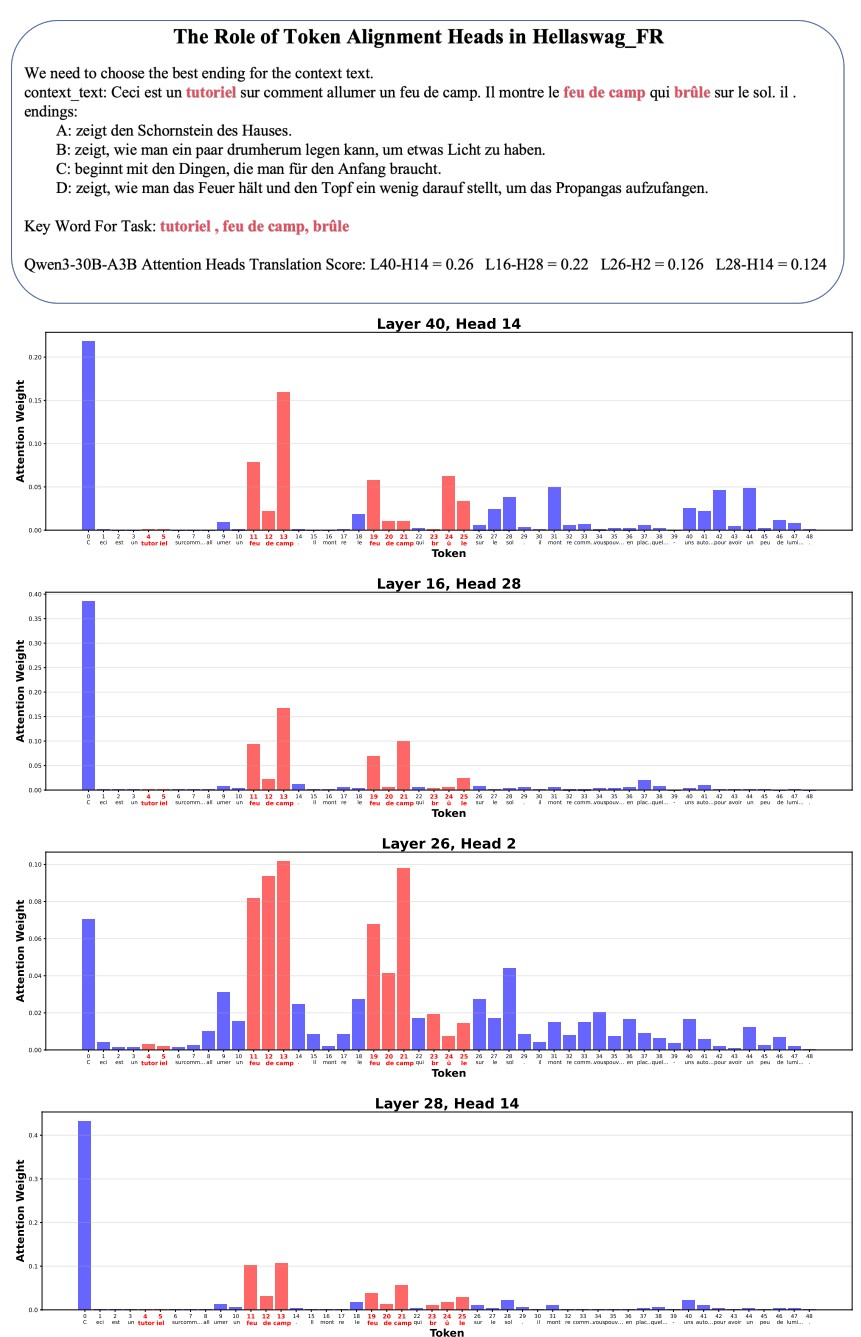

Figure 22: French Hellaswag case. The notation L40-H14 indicates Layer 40, Head 14, where "L" stands for Layer and "H" stands for Head. In this French Hellaswag case, L40-H14, L16-H28, L26-H2 and L28-H14 have translation scores greater than 0.1, identifying them as token alignment heads. We observe that all token alignment heads here can attend to the key tokens.

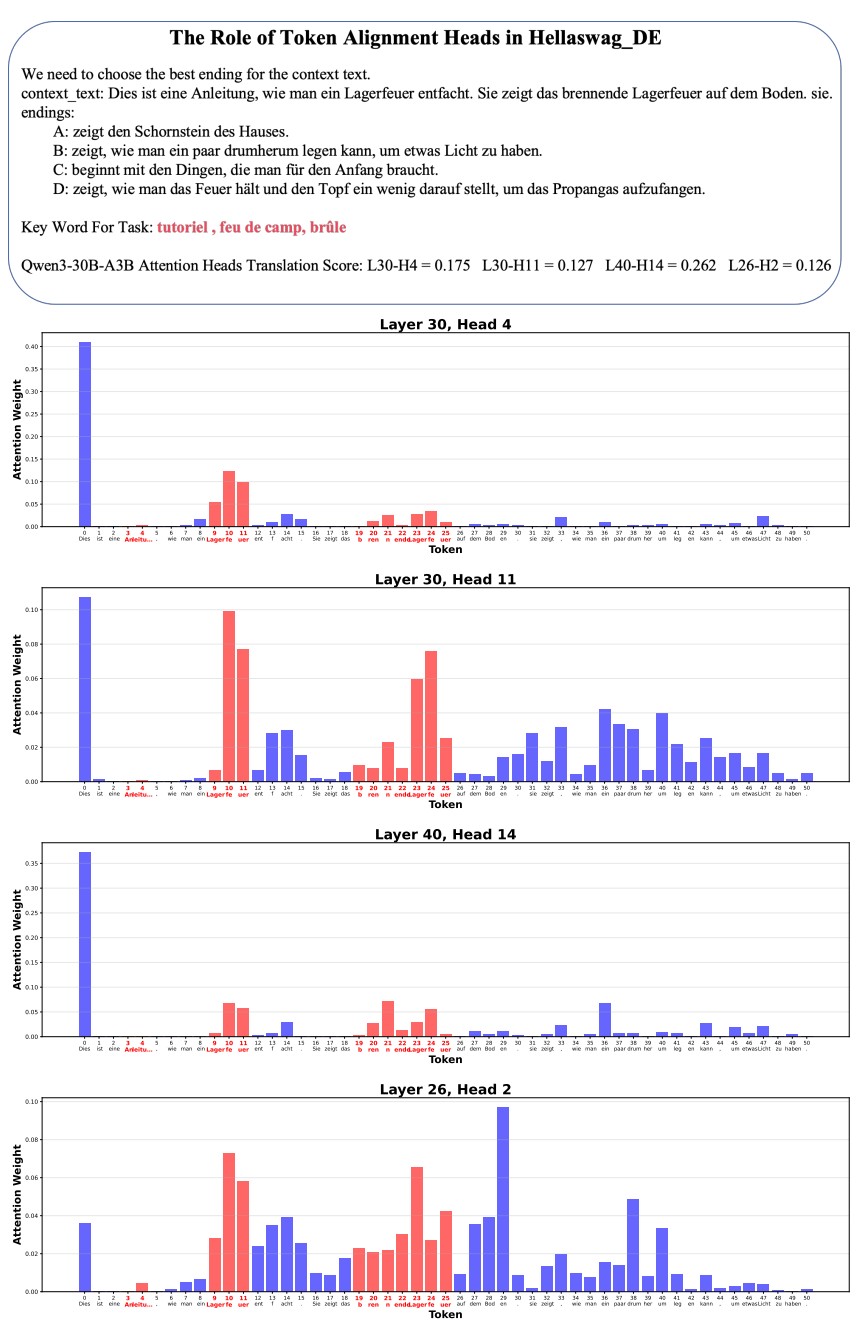

Figure 23: German Hellaswag case. The notation L30-H4 indicates Layer 30, Head 4, where "L" stands for Layer and "H" stands for Head. In this German Hellaswag case, L30-H4, L30-H1, L40-H14 and L26-H2 have translation scores greater than 0.1, identifying them as token alignment heads. We observe that all token alignment heads here can attend to the key tokens.

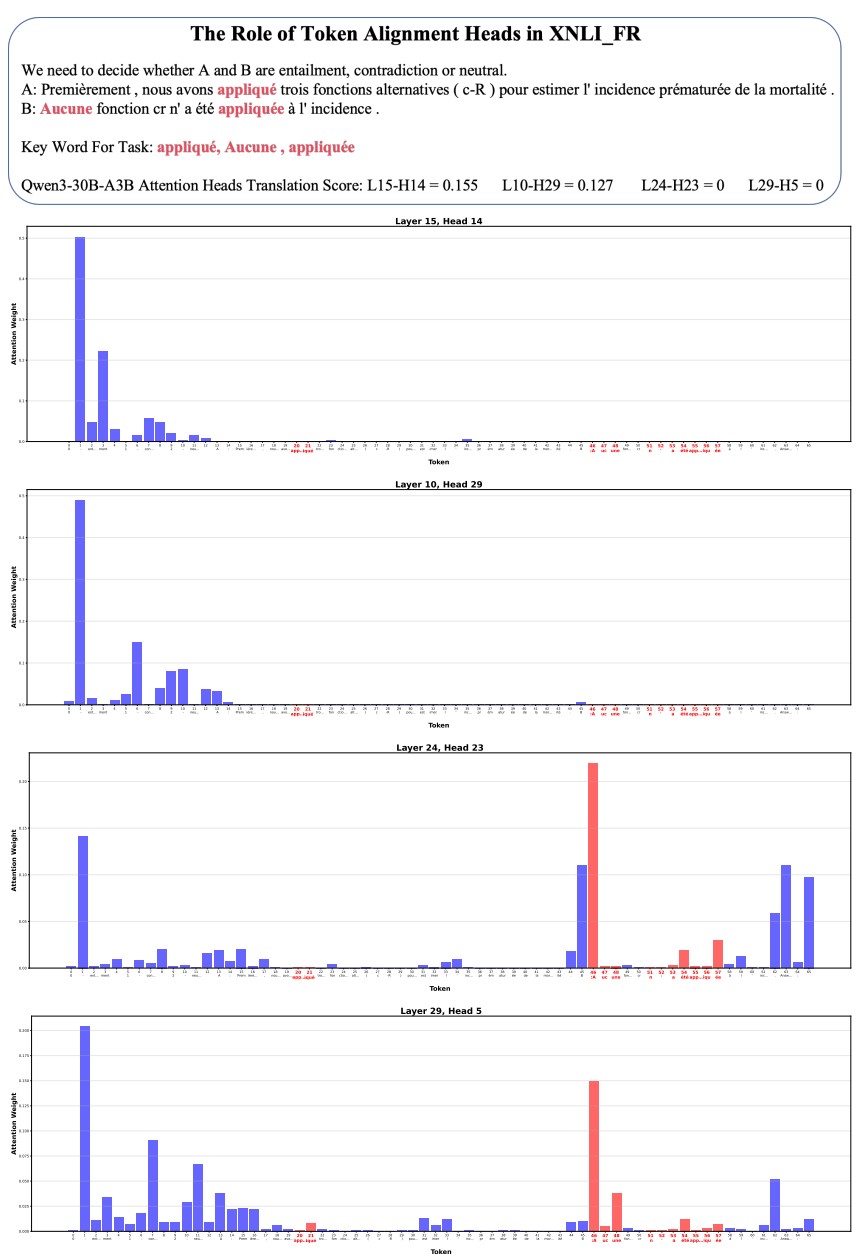

Figure 24: French XNLI case. The notation L15-H14 indicates Layer 15, Head 14, where "L" stands for Layer and "H" stands for Head. In this French XNLI case, Both L15-H14 and L10-H29 have translation scores greater than 0.1, identifying them as token alignment heads, whereas L24-H23 and L29-H5 are not. We observe that the token alignment heads tend to have attention weights close to zero on key tokens, as seen with L15-H14 and L10-H29 in the figure. In contrast, heads with relatively high attention weights on key tokens, such as L24-H23 and L29-H5, have very low translation scores.

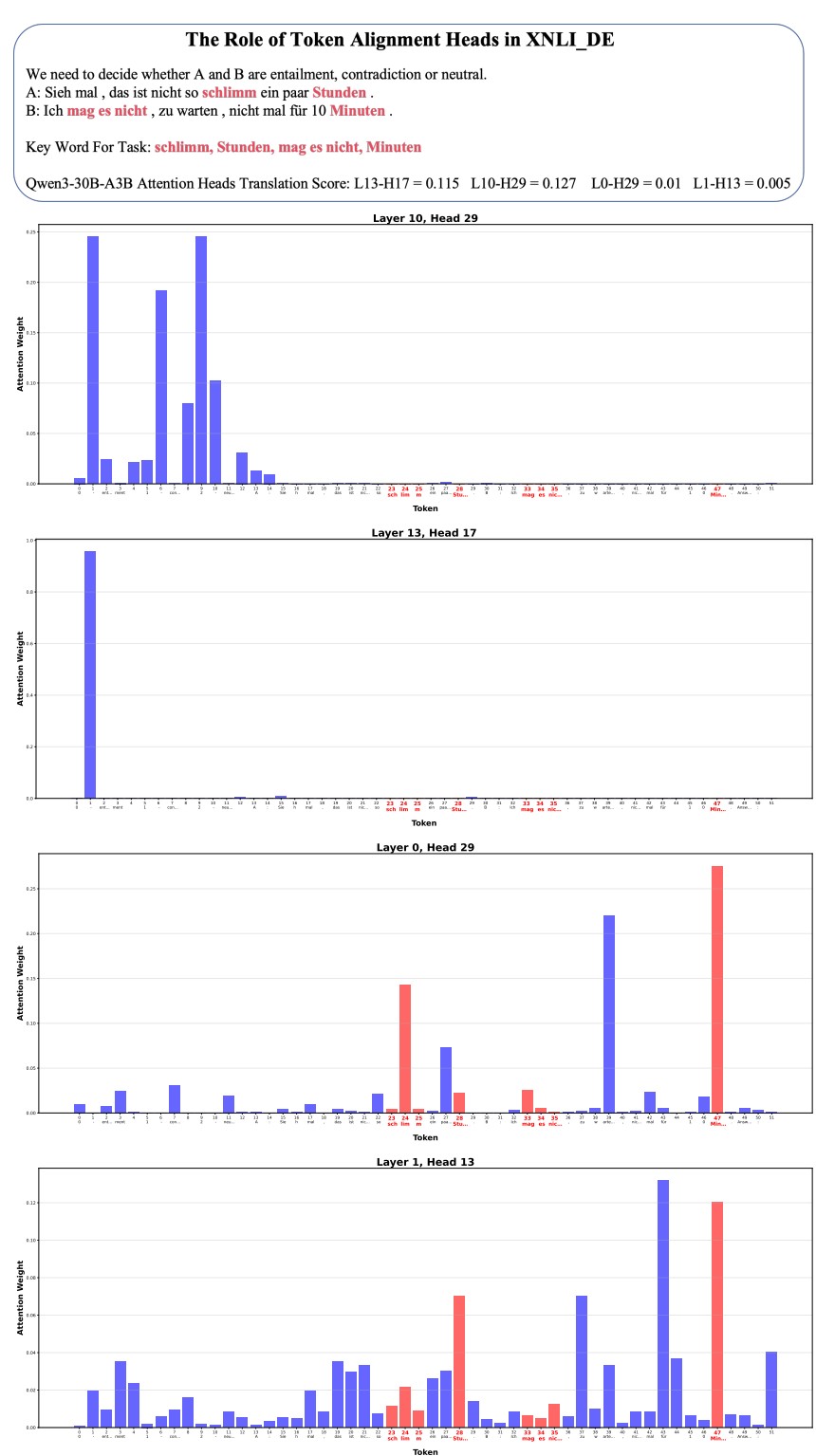

Figure 25: German XNLI case. The notation L13-H17 indicates Layer 13, Head 17, where "L" stands for Layer and "H" stands for Head. In this German XNLI case, Both L13-H17 and L10-H29 have translation scores greater than 0.1, identifying them as token alignment heads, whereas L0-H29 and L1-H13 are not. We observe that the token alignment heads tend to have attention weights close to zero on key tokens, as seen with L13-H17 and L10-H29 in the figure. In contrast, heads with relatively high attention weights on key tokens, such as L0-H29 and L1-H13, have very low translation scores.

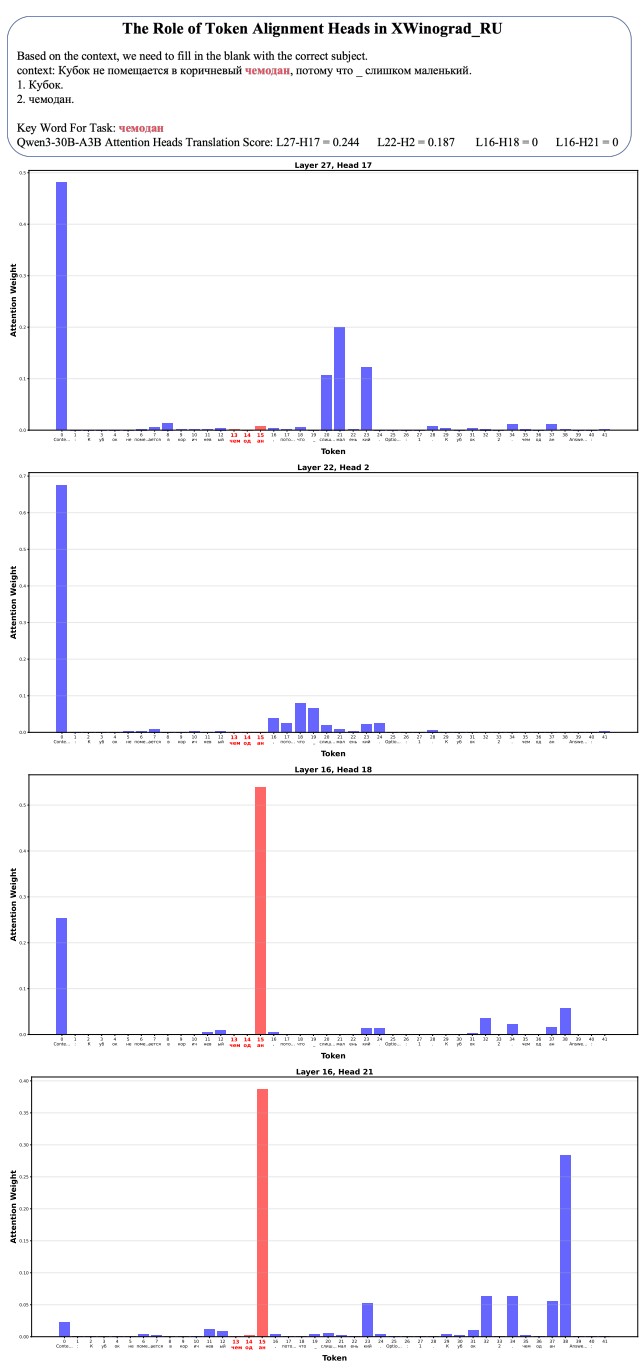

Figure 26: Russian XWinograd case. The notation L27-H17 indicates Layer 27, Head 17, where "L" stands for Layer and "H" stands for Head. In this Russian XWinograd case, Both L27-H17 and L22-H2 have translation scores greater than 0.1, identifying them as token alignment heads, whereas L16-H18 and L16-H21 are not. We observe that the token alignment heads tend to have attention weights close to zero on key tokens, as seen with L27-H17 and L22-H2 in the figure. In contrast, heads with relatively high attention weights on key tokens, such as L16-H18 and L16-H21, have very low translation scores.

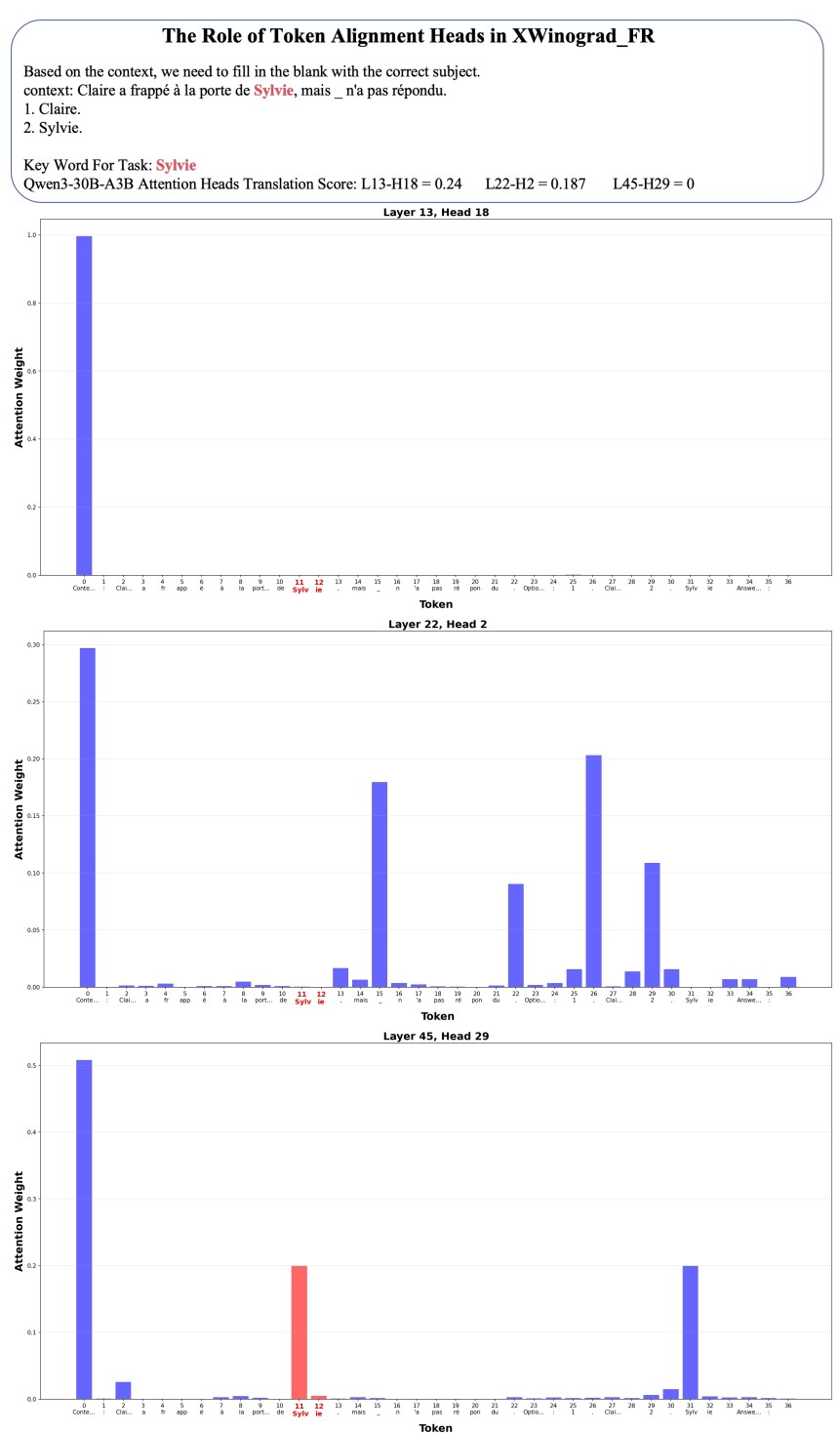

Figure 27: French XWinograd case. The notation L13-H18 indicates Layer 13, Head 18, where "L" stands for Layer and "H" stands for Head. In this French XWinograd case, Both L13-H18 and L22-H2 have translation scores greater than 0.1, identifying them as token alignment heads, whereas L45-H29 is not. We observe that the token alignment heads tend to have attention weights close to zero on key tokens, as seen with L13-H18 and L22-H2 in the figure. In contrast, heads with relatively high attention weights on key tokens, such as L40-H14, have very low translation scores.

## A.8 LANGUAGE CONSISTENCY ACROSS 20 LANGUAGE PAIRS

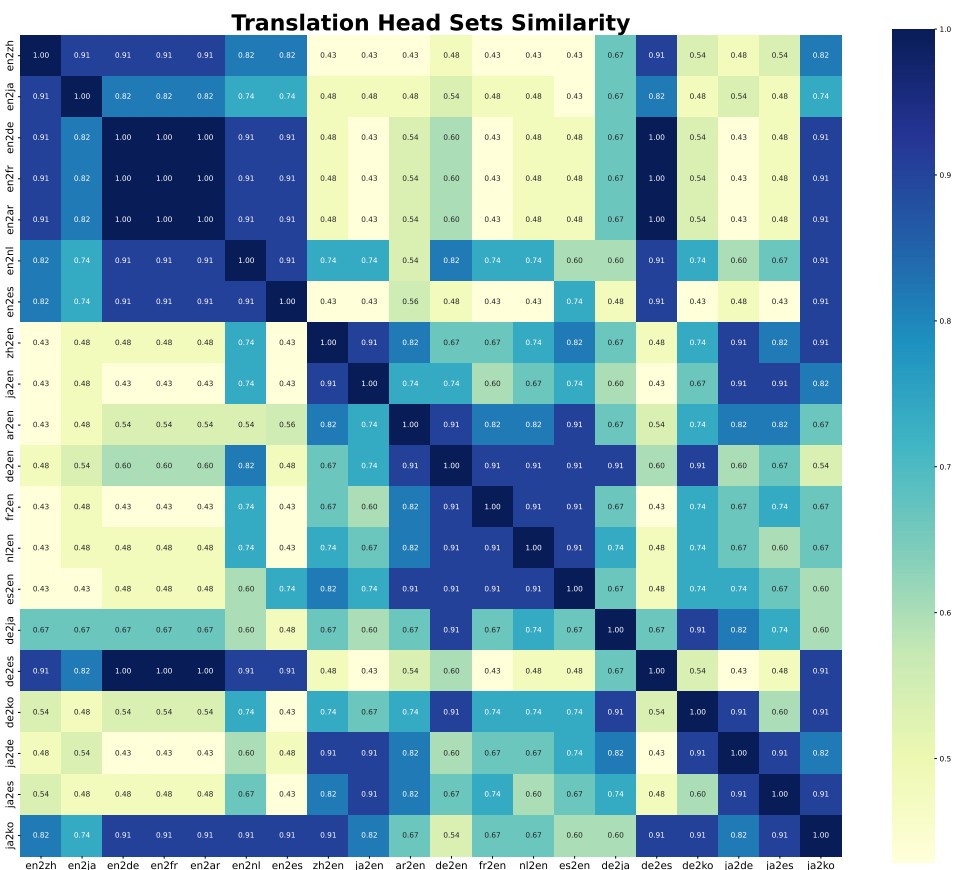

Figure 28: Jaccard similarity matrix of token alignment head sets across 20 language pairs. Here, we broadly categorize the selected language pairs into four groups: en2others (en2zh, en2ja, en2de, en2fr, en2ar, en2nl, en2es), others2en (zh2en, ja2en, ar2en, de2en, fr2en, nl2en, es2en), de2ja/es/ko, and ja2de/es/ko. It is observable that the similarity among the en2others pairs is high, whereas the others2en group and the other language pairs generally exhibit lower similarity.

