# OpenReview forum: "Token Alignment Heads: Unveiling Attention's Role in LLM Multilingual Translation"
_ICLR.cc/2026/Conference — ICLR 2026 Poster_

### Official Review · Reviewer_ZKfU · 2025-10-23

**Soundness:** 3
**Presentation:** 3
**Contribution:** 3
**Rating:** 8
**Confidence:** 4

**Summary:**

The presented paper analyzes that role of specific attention heads for the translation task aka translation heads. These are identified with a word alignment approach and thoroughly analyzed for their location, sparsity, language generalization, emergence, as well as impact on multilingual evaluation benchmarks.

**Strengths:**

- Interesting way of determining translation heads via word alignment properties compared to downstream task performance as done in previous works
- Insightful analysis on the location, sparsity, language generalization, and emergence of translation heads
- Insightful analysis for masking translation heads on downstream tasks including not only translation but also general multilingual benchmarks

**Weaknesses:**

### Weaknesses

- It would've been interesting to analyze the number of translation heads that are specialized for a specific language / language-pair despite them having a large Jaccard index i.e. `de2ja` seems to have a lower similarity to the other language pairs. Generally, it could be that some language pairs are deemed "harder" and need specialized translation heads. I'm a bit surprised to see that `en2de` and `en2nl`, two very closely related language pairs, have the same overlap ratio as `en2de` and `en2zh`, two fairly distant language pairs. It could also be that the deciding factor here is `en` as the source language, rather than the target language.

### Minor Comments

- The "token mapping" is the task of word alignment and has been around for many years in machine translation literature and probably should be mentioned as such.
- All Figures should likely be included as svg graphics instead of low resolution png images since it makes it more appealing to view details. In Figure 3 and Figure 4 some of the labels are way too small to even view properly. A bit more effort on properly scaling the figures would improve the reading experience.

**Questions:**

N/A

---

> ### Author Response · Authors · 2025-11-21
>
> > **Weakness:** It would've been interesting to analyze the number of translation heads that are specialized for a specific language / language-pair despite them having a large Jaccard index … It could also be that the deciding factor here is en as the source language, rather than the target language.
> >
>
> **Response:**  Thank you for this insightful suggestion. Following your comment, we explicitly analyzed specialization patterns of our token alignment heads (TAHs) across language pairs. We find:
> - A small number of TAHs show clear pair-specific behavior. For example, one head is predominantly used for de2ja / de2ko, and another for ja2de / ja2es, suggesting that some “harder” pairs indeed recruit specialized heads.
> - Most TAHs (12 in total) are shared across all examined language pairs, consistent with strong cross-lingual generalization.
> - Several TAHs are shared by multiple, but not all, pairs. Notably, we identify five heads that are used exclusively for en2X directions. This directly supports your hypothesis that sharing the source language (English) is a key factor: it helps explain why the overlap between en2de/en2zh is comparable to that between en2de/en2nl.
>
> To provide a clearer picture, we also extended our analysis from a small set of pairs to 20 language pairs: en2X (en2{zh/ja/de/fr/ar/nl/es}), X2en ({zh/ja/ar/de/fr/nl/es}2en), de2{ja/es/ko}, and ja2{de/es/ko}. The full results are reported in Figure 28 in the Appendix. For example, we now explicitly show that the overlap between de2en and zh2en (0.67) is lower than the high overlap observed for en2de vs. en2zh, further illustrating the asymmetry between sharing a source vs. a target language.
>
>
> > **Minor Comments:** The "token mapping" is the task of word alignment and has been around for many years in machine translation literature and probably should be mentioned as such.
> >
>
> **Response:**  Thank you for pointing this out. We totally agree with your comment. In the revised version, we rename “translation heads” to token alignment heads (TAHs) and add some corresponding work [1,2] into our references.
>
> [1] Peter F. Brown, Vincent J. Della Pietra, Stephen A. Della Pietra, and Robert L. Mercer. 1993. The mathematics of statistical machine translation: parameter estimation. Comput. Linguist. 19, 2 (June 1993), 263–311.
>
> [2] Franz Josef Och and Hermann Ney. 2003. A Systematic Comparison of Various Statistical Alignment Models. Computational Linguistics, 29(1):19–51.
>
>
>
> > **Minor Comments:** All Figures should likely be included as svg graphics instead of low resolution png images since it makes it more appealing to view details. In Figure 3 and Figure 4 some of the labels are way too small to even view properly. A bit more effort on properly scaling the figures would improve the reading experience.
> >
>
> **Response:**  Thank you for your suggestion. In the revised manuscript, we have adjusted the font size of all figures and converted them to a high-resolution pdf format.

---

### Official Review · Reviewer_DWDG · 2025-10-29

**Soundness:** 2
**Presentation:** 2
**Contribution:** 1
**Rating:** 4
**Confidence:** 4

**Summary:**

This paper utilizes token-aligned attention scores to detect salient translation heads (TH) in LLMs. The authors find that such attention heads significantly affect multilingual proficiency, as validated by the multilingual dataset FLORES-101. Experiments demonstrate substantial overlapping of salient heads across 10 different language pairs and confirm the deterministic effect of these heads through comprehensive ablation studies.

**Strengths:**

1. This work presents a token-aligned detection approach with unsupervised LLM labelling that requires inference only. It demonstrates the significance of THs through a masking approach for machine translation.
2. This work illustrates the properties of THs through detailed analysis across different models and languages.
3. This work unveils the learning process for translation heads during model training.

**Weaknesses:**

1. This work's organization lacks rigorous experimental settings. For instance, Figure 1 fails to adequately demonstrate the function of THs through its single example. Additionally, the selection of models appears arbitrary, lacking consistency in either model size or type.
2. The figures throughout the paper are blurry, making it difficult to discern key numerical values (particularly in Figures 3 and 4).
3. The proposed data augmentation method does not demonstrate significant improvements in Table 1 when compared to the baselines.
4. Despite numerous recent studies exploring the internal mechanisms of multilingual capacity in LLMs, this work lacks a thorough review of related work in this field.

**Questions:**

1. Given that this is not the first study to explore the multilingual abilities of LLMs, what are the key distinctions between this work and previous research?
2. How do current machine translation metrics demonstrate the significance of translation heads on overall MT performance?
3. Since BLEU and chrF++ metrics cannot inherently recognize languages, is it possible that these translation heads are not fully responsible for multilingual capacity if LLMs cannot function properly without them?

---

> ### Author Response · Authors · 2025-11-21
>
> We thank the reviewer for the careful reading of our paper and for the constructive comments. We carefully respond to each point in turn and describe the additional analyses and revisions we have made in the updated pdf revision.
>
> > **Weakness 1:** This work's organization lacks rigorous experimental settings. For instance, Figure 1 fails to adequately demonstrate the function of THs through its single example. Additionally, the selection of models appears arbitrary, lacking consistency in either model size or type.
> >
>
> **Response:** We appreciate this comment and and we have revised the paper accordingly. In the revised draft, we have presented more representative examples in Appendix A.4. Empirically, we observe three dominant error types after ablating the top translation heads:
> 1. **No Translation (46%)**: the model fails to produce the target language and instead copies the source text. This is the type illustrated in the revised pdf's Figure 10. In these cases, once THs are masked, the model can no longer perform cross-lingual token alignment for certain queries and thus does not produce target-language output at all.
> 2. **Missing Details (36%)**: the output is in the correct target language, but key details in the source are no longer aligned and therefore omitted, leading to incomplete or under-specified translations. See Figure 11 for such cases of the revision pdf.
> 3. **Hallucinated (18%)**: the model produces target-language content that is not supported by the source. We hypothesize that, after alignment is disrupted, the model falls back to generating semantically “nearby” target tokens without grounding them in the source. Cases are shown in Figure 12 in the revision pdf.
>
> **Regarding choice of models.**
>
>  To make the work more rigorous, in the updated experiments, we provide analysis across **four major multilingual LLM families** and a broad range of sizes (**adding Gemma2 series, Qwen3-8B/14B/32B, Mixtral 8x7B, LLaMA3.1-1B in the updated manuscript**):
>   - **Llama3**: 1B / 8B (base and instruct)
>   - **Gemma2**: 2B / 9B / 27B (base and instruct)
>   - **Mistral**: 7B (base and instruct)
>   - **Qwen3**: 1.7B / 8B / 14B (base) and 1.7B / 8B / 14B / 32B (instruct)
>
> In addition, we include two **MoE** models, **Mixtral-8×7B** and **Qwen3-30B-A3B**, to cover non-dense architectures. Across all of these models, we observe the same qualitative patterns:
> - A small, sparse subset of heads with high translation scores emerges;
> - Ablating them causes large drops in translation performance, whereas masking an equal number of random heads has minimal effect.
>
> We  summarize this expanded model set and the corresponding results in the revised manuscript (Appendix A.5–A.6). We hope that these additional analyses and clarifications address the reviewer’s concerns about experimental rigor. We are pleased to have more analysis if there are any concerns.
>
>
> > **Weakness 2:** The figures throughout the paper are blurry, making it difficult to discern key numerical values (particularly in Figures 3 and 4).
> >
>
> **Response:** We thank the reviewer for pointing this out. In the revised manuscript, we have regenerated all figures with larger font sizes for all text and numerical labels, and exported them as high-resolution vector (PDF) figures. This eliminates the blurriness in Figures 3 and 4 and makes the key values clearly readable, even when zoomed in. If there are still some dissatisfied figures, just let us know and we will be more than happy to improve them.

---

> > ### Author Response · Authors · 2025-11-21
> >
> > > **Weakness 3:** The proposed data augmentation method does not demonstrate significant improvements in Table 1 when compared to the baselines.
> > >
> >
> > **Response:**  We appreciate the reviewer’s comment and would like to clarify the setup and highlight additional results that we have added in the revised version.
> >
> > First, the translation-rich data selected by TRater in “enhance” setting only accounts for 1.68% of the training corpus. Under such a small data ratio, we view the observed gains as meaningful: for example, for 1.5B model, on FLORES (chrF++) the score improves from 43.87 → 46.68 (+2.81). This suggests that the selected subset is highly influential for translation quality. Furthermore, we also observed improvements across other benchmarks, including Hellaswag_ML, ARC_C_ML, ARC_E_ML and XMMLU.
> >
> > To more directly address the concern about the magnitude and robustness of improvements, we have conducted two additional experiments:
> > 1. Larger model (8B, 1T tokens, 1.68% selected data):
> >
> > |            | flores_chrf | MMMLU  | hellaswag_ml | arc_c_ml | arc_e_ml | xwinograd | xstorycloze | xnli   | xcopa  | xmmmlu  |
> > |-----------|-------------|--------|--------------|----------|----------|-----------|-------------|--------|--------|---------|
> > | **baseline** | 50.11      | **32.12** | 55.28       | 40.54   | 65.34   | **81.16** | 63.58      | 44.26 | **70.13** | 37.36  |
> > | **remove**   | 49.38      | 31.48 | 55.03       | 40.48   | 65.18   | 80.55     | 63.48      | 43.46 | 69.9   | 37.225 |
> > | **enhance**  | **53.52**  | 31.87 | **56.01**   | **40.95** | **65.97** | 80.89   | **63.89**  | **44.26** | 69.96 | **38.001** |
> >
> >   In this setting, our method yields obvious gains over the baseline: FLORES:**+3.41**, Hellaswag_ML:**+0.73%**, ARC_C_ML:**+0.41%**, ARC_E_ML:**+0.63%**, XMMLU:**+0.64%**
> >
> > 2. Larger data ratio (1.5B model, 1T tokens, 5% selected data):
> >
> > |               | flores_chrf | MMMLU  | hellaswag_ml | arc_c_ml | arc_e_ml | xwinograd | xstorycloze | xnli   | xcopa  | xmmlu   |
> > |--------------|-------------|--------|--------------|----------|----------|-----------|-------------|--------|--------|---------|
> > | **baseline**     | 43.87      | **26.89** | 44.88       | 31.4    | 53.97   | **75.7**  | 58.4       | 41.94 | 62.88 | 30.712  |
> > | **remove**       | 41.33      | 26.58 | 44.69       | 31.37   | 54.54   | 73.63     | 58.15      | **42.1** | 63.2  | 30.844  |
> > | **enhance_1pct** | 46.68      | 26.71 | 44.95       | 31.54   | 54.94   | 74.33     | **58.44**  | 41.72 | 63.7  | 30.884  |
> > | **enhance_5pct** | **47.54**  | 26.53 | **45.27**   | **32.19** | **55.9** | 74.84   | 58.19      | 42.09 | **64.13** | **31.338** |
> >
> > When we increase the proportion of selected data to 5%, the improvements become even more pronounced: FLORES:**+3.67**, Hellaswag_ML:**+0.39%**, ARC_C_ML:**+0.79%**, ARC_E_ML:**+1.93%**, XMMLU:**+0.626%**
> >
> > Taken together, these results show that even when the selected data constitutes a very small fraction of the corpus, our Translation Head (which we more precisely refer to as Token Alignment Heads, TAH, in the revised manuscript)–guided augmentation leads to consistent and non-trivial gains across multiple multilingual benchmarks. We will include the full tables in the appendix of the revised version.

---

> ### Author Response · Authors · 2025-11-21
>
> > **Weakness 3:** Despite numerous recent studies exploring the internal mechanisms of multilingual capacity in LLMs, this work lacks a thorough review of related work in this field.”
> >
>
> >**Question 1:** Given that this is not the first study to explore the multilingual abilities of LLMs, what are the key distinctions between this work and previous research?
> >
>
>
> **Response:**  We thank the reviewer for the constructive feedback. Based on our investigation, there are four main lines of research in this literature:
>
> 1. Semantic-space
>  Prior work studies the geometry of multilingual representations and often concludes that models “think in English” or in a shared latent semantic space in middle layers (e.g., Wendler et al., 2024; Schut et al., 2025; Zhao et al., 2024; Wu et al., 2024; Harrasse et al., 2025). These works explain where multilingual information lives and in what form it is represented (English or Language-Agnostic Space). Our results are complementary: we identify translation heads concentrated in similar middle layers and show that they implement token-level cross-lingual alignment, routing the aligned source token’s representation into the target position.
> 2. Neuron-level mechanisms
>  Another line of work finds neurons that encode language identity or language-specific features and shows that ablating or fine-tuning them selectively affects particular languages (Tang et al., 2024; Zhao et al., 2024). This explains which subcircuits are responsible for “being in language X.” In contrast, we operate at the head level and focus on the cross-lingual mapping step (source token to aligned target token), i.e., how information moves between languages during translation.
> 3. Head-level mechanisms
>  Recent work identifies “language heads” or translation-related heads by ranking heads via their impact on downstream loss, perplexity, or logits on specific benchmarks, sometimes using path patching (e.g., Liu et al., 2025; Zhang et al., 2025). Our approach is closely related but uses a different identification signal: we define Translation Heads (TH) using an alignment-based translation score on parallel corpora—heads are selected because they consistently link target tokens to their externally aligned source tokens, independent of any particular evaluation task. Masking experiments are then used only as a causal validation step. This makes our notion of specialization explicitly lexical and cross-lingual.
> 4. Mechanistic interpretability and multilingual circuits.
>  Work on induction heads, retrieval heads, and circuits (Elhage et al., 2021; Olsson et al., 2022; Anthropic, 2024; Wu et al., 2025; Long, 2025; Zhang et al., 2024) shows that a small number of specialized heads can mechanistically explain non-trivial capabilities such as in-context learning or long-context retrieval. Our study follows the same “mechanism-first + intervention” paradigm but in the multilingual setting: we identify a sparse subcircuit, a.k.a translation head, analyze its behavior over pre-training, and ask whether it is reused across languages and models.
>
> Addressing the reviewer’s question about key distinctions, our paper is, to our knowledge, the first to:
> - define translation heads using an alignment-grounded translation score rather than task loss or logit changes alone;
> - show that these alignment-defined heads are also the ones with the strongest causal impact on MT performance;
> - characterize their universality, sparsity, and training dynamics across diverse LLMs (dense/MoE, base/instruct, 1.7B–30B) and tens of language pairs;
> - demonstrate a practical application of this mechanism by using the resulting TRater score for multilingual data selection.
> We have added these connections and distinctions to the revised version. And we sincerely hope that the discussion could resolve the reviewer’s concern and clarify how our work advances the understanding of internal multilingual mechanisms beyond existing studies.
>
> reference of the related work can be found in the revised manuscript

---

> > ### Author Response · Authors · 2025-11-21
> >
> > Thank you for your thoughtful review and feedback on our submission. We appreciate the time and effort you invested in understanding our work. We would like to address the concerns you raised:
> >
> > > **Q2:** How do current machine translation metrics demonstrate the significance of translation heads on overall MT performance?
> > >
> >
> > **Response:**  We appreciate the reviewer’s question about how standard MT metrics demonstrate the significance of translation heads for overall translation quality.
> >
> > In the main paper, we quantify the impact of masking translation heads primarily using corpus-level **BLEU** and **chrF++**. Across multiple language pairs and models, ablating the top-k TAHs leads to substantial drops in both metrics, while masking the same number of random heads has only minor effects. This gap between “TH ablation” and “random ablation” is what we interpret as evidence that TAHs form a sparse, specialized subcircuit that is crucial for MT performance.
> >
> > We agree, however, that BLEU and chrF++ are based on n-gram or character overlap and only partially capture semantic adequacy and fluency. To address this limitation, in the revised version we additionally evaluate BLEURT and COMET under the same ablation settings, and report the results in Appendix A.6 (Figures 19, 20, and 21). These learned metrics are designed to be more sensitive to meaning preservation and overall translation quality.
> >
> > The new results show that:
> > - When we mask the top-k translation heads, **BLEURT** and **COMET** scores consistently degrade across language pairs and models.
> > - The degradation pattern closely mirrors what we observe for **BLEU** and **chrF++**: ablations targeting THs cause much larger drops than ablations of random heads with the same cardinality.
> >
> > Taken together, the agreement between surface-form metrics (BLEU/chrF++) and semantic metrics (BLEURT/COMET) supports our claim that translation heads are not only important at the character/n-gram level, but also play a central role in maintaining the semantic quality of translations according to current MT evaluation standards.

---

> > > ### Author Response · Authors · 2025-11-21
> > >
> > > > **Q3:** Since BLEU and chrF++ metrics cannot inherently recognize languages, is it possible that these translation heads are not fully responsible for multilingual capacity if LLMs cannot function properly without them?
> > > >
> > >
> > > **Response:**  We appreciate this concern and would like to clarify both our metric choices and the scope of our claims.
> > >
> > > First, we agree that BLEU and chrF++ are surface-form metrics that operate at the character or n-gram level and do not explicitly “recognize” languages. As noted above, to mitigate this limitation we also evaluate BLEURT and COMET, which are learned metrics designed to capture semantic adequacy and fluency.
> > >
> > > Second, our paper does **not** claim that translation heads are “fully responsible” for the model’s multilingual capacity. We make a narrower claim: TAHs form a sparse subcircuit that is crucial for cross-lingual token alignment. Consistent with this, we observe that on **translation-focused benchmarks** (e.g., FLORES) ablating TAHs causes much larger drops than ablating random heads, whereas on **other multilingual tasks** such as **Hellaswag_ML**, **ARC_C_ML**, and **ARC_E_ML** the degradation is noticeably smaller. For tasks such as **XWinograd** and **XNLI**, the effect of TH ablation is often comparable to or even less than that of random-head ablation, and the drops are small in absolute terms. In other words, even when TAHs are masked, the model largely retains its performance on reasoning or knowledge-heavy multilingual benchmarks, suggesting that those abilities rely on different circuits than the ones we identify.
> > >
> > > Finally, our qualitative error analysis further supports this view. In Appendix A.3 we categorize the errors that occur when masking THs into three types: (i) **cannot translate** (46%), where the model copies the source text rather than producing the target language; (ii) **detail missing** (36%), where the output is in the correct target language but omits important content; and (iii) **hallucinated content** (18%), where the model generates target-language material not supported by the source. Thus, for 54% of the affected examples, the model still outputs in the target language after TAH ablation but with degraded adequacy or faithfulness. This indicates that ablating TAHs does not make the LLM “unable to function” in other languages in general; instead, it specifically disrupts the **token-level alignment and faithful information transfer** that underlie high-quality machine translation.
> > >
> > > Taken together, these results support a more nuanced conclusion: translation heads are not the sole locus of multilingual capacity, but they are a specialized and causally important component for lexical translation, while other multilingual abilities (e.g., reasoning, world knowledge) are supported by different parts of the network.

---

> > > > ### Comment · Reviewer_DWDG · 2025-11-22
> > > >
> > > > Thanks for your response,
> > > >
> > > > - The authors solve the main concerns for this paper. I will adjust the scores to be more positive, as they clarify the value of this paper, while I still have several minor suggestions.
> > > >   - The first is the contribution part, one main part is the function vector for multilingual ability. To my best knowledge, the function vector (https://arxiv.org/abs/2310.15213) and the translation-head vector (https://arxiv.org/abs/2410.07054) are the two main works.
> > > >   - The data augmentation experiments show a marginal effect on the final performance, while it is an optional choice to utilize the findings on translation heads. I suggest modifying the corresponding contribution part.
> > > >   - After reading your latest draft, it is better to put the location of the related work part before the conclusion. The missing information in the related work is suggested to be addressed by reorganizing the paper's structure.

---

> > > > > ### Author Response · Authors · 2025-11-22
> > > > >
> > > > > Thank you very much for the follow-up and for updating your score. We are glad that our previous responses addressed your main concerns. We have revised the manuscript according to your new suggestions, as summarized below:
> > > > >
> > > > > 1. **Location and content of the related work section.**
> > > > >  We have moved the Related Work section to appear before the Conclusion section. In addition, we expanded it with a dedicated paragraph discussing the main prior works on function vectors and translation-head vectors that you mentioned, and clarified how our token-alignment heads connect to this line of research.
> > > > >
> > > > > 2. **Wording around the data augmentation experiments.**
> > > > >  As suggested, we have softened the phrasing regarding the data augmentation experiments. In particular, we removed claims of “significant improvement” in the abstract and experimental discussion and now describe the gains in a more cautious way.
> > > > >
> > > > > We sincerely appreciate your constructive comments and the time you spent reviewing our work.

---

### Official Review · Reviewer_nLP1 · 2025-10-30

**Soundness:** 3
**Presentation:** 3
**Contribution:** 3
**Rating:** 6
**Confidence:** 4

**Summary:**

The paper investigates the role of translation heads in translation and multilingual tasks. Translation heads are identified based on their accuracy in token mapping. The authors identify several features of these translation heads, including i) that they are universal in all LLMs, ii) that they only a few translation heads needed, iii) that the translation heads are consistent across language pairs, iv) translation capability is severely impacted when these heads are masked and v) that they occasionally have an impact of other multilingual tasks. Finally, they trace the development of these heads across cycles of pretraining and show how the attention heads can be used to filter multilingual data for more efficient training.

**Strengths:**

1. The idea that cross-lingual word-mapping heads exist in multilingual LLMs is interesting, and how they demonstrate impact on translation and some multilingual tasks.
2. The experiments that evaluate key features of the heads are thorough and comprehensive.
3. The writing is clear. All motivations and experimental designs are easy to understand.

**Weaknesses:**

1. Translation heads are identified solely through cross-lingual one-to-one token mapping. However, translation as a process involves more than token mapping alone, e.g., resolving morphological ambiguity. Current name that is translation heads sounds a little misleading.
2. Previous literature show that transformer-based models tend to process multilingual input in English in the middle layers:
- https://aclanthology.org/2024.acl-long.820/
- https://arxiv.org/abs/2411.04986
- https://arxiv.org/abs/2402.18815
This seems incompatible with your finding in Figure 4 where translation heads are mostly gathered in the middle. What roles exactly do translation heads in the middle layer play in solving multilingual queries?
3. While the role of translation heads is clear on translation task, the results on other multilingual task are less conclusive. There is no clear analysis leading us to understand why this is the case, or the kind of role the attention heads play in solving these tasks.
4. Minor: fonts are too small for all figures.

**Questions:**

1. Can you reconcile your finding in Figure 4 with prior literature? see Weakness 2.
2. In some cases (Figure 9, xwinograd, xnli), pruning translation heads seem to give better results, why?
3. What is L in Equation 4?

---

> ### Author Response · Authors · 2025-11-21
>
> We thank the reviewer for the thoughtful and detailed review, as well as for the positive assessment of our idea, experimental thoroughness, and clarity of writing. Below, we respond to each weakness and question in turn and describe the clarifications and revisions we have made in the updated manuscript.
>
> > **Weakness 1:** Translation heads are identified solely through cross-lingual one-to-one token mapping. However, translation as a process involves more than token mapping alone, e.g., resolving morphological ambiguity. Current name that is translation heads sounds a little misleading.
> >
>
> **Response:** We agree that translation goes well beyond one-to-one token mapping. Our aim is not to claim that a small set of attention heads implements the whole translation process, but rather to isolate and study one specific component: cross-lingual token alignment. To make this scope explicit and avoid any overclaiming, we will rename these heads as token alignment heads (TAHs) and use this terminology consistently in the revised paper.
>
>
> > **Weakness 2:** Previous literature show that transformer-based models tend to process multilingual input in English in the middle layers… This seems incompatible with your finding in Figure 4 where translation heads are mostly gathered in the middle. What roles exactly do translation heads in the middle layer play in solving multilingual queries?
> >
> > **Question 1:** Can you reconcile your finding in Figure 4 with prior literature? see Weakness 2.
> >
>
> **Response:**  Thank you for your insightful comment. We believe our finding in Figure 4 that translation heads (now Token Alignment Heads, TAHs) are mostly concentrated in the middle layers is  compatible with the previous work suggesting an "English-centric" processing space in multilingual LLMs. The reasons are as follows: As the studies cited by the reviewer (and others, such as Anthropic's work on attribution https://transformer-circuits.pub/2025/attribution-graphs/biology.html) have shown, the middle layers of Transformer models develop language-agnostic representations. This means that tokens with similar semantics from different languages are mapped to nearby points in the model's hidden state space. This proximity enables attention heads to  perform token alignment conveniently when dealing with translation tasks. Thus, concentration in the middle layers is a direct consequence of this language-agnostic space, as this is the ideal layer position to align concepts before further processing.
>
> Furthermore, for the role of TAH in dealing with multilingual queries, we hypothesize that the TAHs might behave as a "Bridge" to the English-Centric Space. Specifically, according to Zhao et al. (24), while the middle-layer representations are becoming language-agnostic, the Feed-Forward Networks (FFNs) in these same layers still retain vast amounts of multilingual knowledge. The token alignment heads (TAHs) are instrumental in connecting this stored multilingual knowledge to the English-centric space. They align non-English representations to their English counterparts, enabling the model to leverage its powerful reasoning capabilities, which are often most developed in the English space. We view a more direct causal test of this hypothesis as an interesting direction for future work.
>
> Y. Zhao, W. Zhang, G. Chen, K. Kawaguchi, L. Bing. 2024. How Do Large Language Models Handle Multilingualism, NeruIPS, 2024

---

> ### Author Response · Authors · 2025-11-21
>
> > **Weakness 3:** While the role of translation heads is clear on translation task, the results on other multilingual task are less conclusive. There is no clear analysis leading us to understand why this is the case, or the kind of role the attention heads play in solving these tasks.
> >
>
> **Response:** In Section 5.2, and further in the revised Appendix A.7, we explicitly distinguish two classes of multilingual tasks and observe different behaviors of token alignment heads (TAHs).
>
> 1. Tasks that still depend on cross-lingual access to knowledge.
> Benchmarks such as Hellaswag_ML, ARC_C_ML, and ARC_E_ML require the model to combine multilingual common-sense or scientific knowledge with some degree of cross-lingual understanding. On these tasks, masking TAHs leads to noticeably larger drops than masking the same number of random heads, though the effect is smaller than on Flores. This is consistent with our “bridge” view from Weakness 2: TAHs help route information encoded in non-English tokens into the English-centric shared space where much of the model’s knowledge and reasoning circuitry resides. To make this more concrete, in the revised version we add qualitative analyses in Appendix A.7. For example, on Hellaswag_ML we find TAHs that reliably attend to semantically critical source tokens.
>
> 2. Tasks that are largely translation-independent.
> In contrast, tasks such as XWinograd and XNLI are more heavily driven by logical reasoning, coreference resolution, or high-level semantics, and much less by explicit translation or cross-lingual token alignment. Empirically, we observe that:
>   - Ablating TAHs has relatively small effects on these benchmarks, sometimes even smaller than ablating random heads
>   - In our attention visualizations (Appendix A.7), most TAHs place near-zero attention on the key tokens.
> This suggests that the circuits solving these tasks rely primarily on other heads and MLPs rather than on the token-alignment subcircuit we identify.
>
>
> > **Weakness 4:** Minor: fonts are too small for all figures.
> >
>
> **Response:** We have updated the figures in the revised version. Thank you for the comment.
>
>
> > **Question 2:** In some cases (Figure 9, xwinograd, xnli), pruning translation heads seem to give better results, why?
> >
>
> **Response:** Thank you for pointing this out. Actually, the effect appears only in a specific setting on the large Qwen3-30B-A3B and is small in magnitude. We interpret it as a consequence of (1) the very large number of heads in this model and (2) the fact that these tasks rely only weakly on token-level cross-lingual alignment.
>
> First, Qwen3-30B-A3B has many attention heads, and consequently our procedure identifies close to 90 token alignment heads (TAHs). For translation-focused benchmarks (e.g., FLORES), masking these TAHs causes clear performance drops, as discussed in the main paper. However, for XWinograd and XNLI, which are dominated by logical reasoning rather than explicit translation, our qualitative analysis (Appendix A.7) shows that most TAHs place near-zero attention on key tokens  and tend to attend to context tokens that are largely irrelevant to the label. In other words, for these tasks, many TAHs behave more like noise than useful signals.
>
> Second, when we vary the number of masked TAHs on Qwen3-30B-A3B, we see a non-monotonic pattern rather than “the more pruning the better.” Concretely, pruning a moderate number of TAHs yields small improvements over the base model on XWinograd and XNLI, but too much pruning eventually hurts:XWinograd improves slightly from 82.87 to 83.61 when masking around 40\~50 TAHs, then declines as more heads are removed (down to 82.53 when all \~90 TAHs are masked). And XNLI similarly peaks around 45.66 when masking 50 TAHs, but drops back below the baseline (44.0) when all TAHs are removed. We believe this phenomenon is consistent with prior work on attention head redundancy (Michel P, et al. 19), which shows that removing some weak or noisy heads can act as a mild regularizer and sometimes marginally improve performance, especially in over-parameterized settings.
>
> | Model             | xwinograd | xnli |
> |-------------------|-----------|---------|
> | **qwen3-30B-base**  | 82.87     | 44.04   |
> | **qwen3-30B-top5**  | 83.07     | 44.86   |
> | **qwen3-30B-top10** | 83.20     | 44.93   |
> | **qwen3-30B-top20** | 83.56     | 45.19   |
> | **qwen3-30B-top30** | 83.50     | 45.30   |
> | **qwen3-30B-top40** | 83.61 | 45.41   |
> | **qwen3-30B-top50** | 83.42     | 45.66 |
> | **qwen3-30B-top60** | 83.31     | 45.38   |
> | **qwen3-30B-top70** | 83.23     | 44.85   |
> | **qwen3-30B-top80** | 83.04     | 44.37   |
> | **qwen3-30B-top90** | 82.53     | 43.87   |
>
> Michel P, Levy O, Neubig G. Are sixteen heads really better than one?. Advances in neural information processing systems, 2019

---

> > ### Author Response · Authors · 2025-11-21
> >
> > > **Question 3:** What is L in Equation 4?
> > >
> >
> > **Response:** L is the token level cross entropy loss. We have clarified it in the revised version.

---

### Author Response · Authors · 2025-12-02
**Summary of Reviews and Author Response**

Dear Area Chair and Reviewers,

Thank you very much for the time and care you devoted to evaluating our submission. Your comments substantially helped us refine and strengthen the paper. For convenience, we briefly summarize below how the revised manuscript and rebuttal address the main points raised in the reviews.

Reviewers highlighted the novelty of our analysis of multilingual LLMs through **Token Alignment Heads (TAHs)**, which make concrete the token-level alignment behavior underlying translation and connect low-level attention patterns with high-level multilingual performance. They also noted the systematic analysis of head properties (universality, sparsity, emergence), the breadth of models evaluated, and the overall clarity of the presentation.

To address the key concerns, we have made the following clarifications and additions in the revised version:

1. **Clarification of scope and terminology (Reviewers nLP1, ZKfU)**

    In line with Reviewer nLP1’s suggestion, we renamed the originally termed “Translation Heads” to Token Alignment Heads (TAHs) throughout the paper to emphasize that they implement cross-lingual token mapping (word alignment, as highlighted by Reviewer ZKfU) rather than full translation. This avoids overclaiming and makes the scope explicit.

2. **Deepened mechanistic analysis (Reviewer nLP1)**
    - **Layer distribution.** We clarified why TAHs concentrate in middle layers, relating this to prior work showing that middle layers encode language-agnostic semantic representations. Because semantically similar tokens are already close in the hidden-state space at these layers, they form a natural locus for token alignment operations.
     - **“Bridge” hypothesis and task-specific roles.** We proposed and illustrated the Bridge Hypothesis: TAHs connect multilingual knowledge stored in middle-layer FFNs to the predominantly English-centric reasoning circuitry. This framework explains the divergent effects observed across multilingual tasks, which we further support with qualitative case studies in Appendix A.7:
       - Knowledge-dependent tasks (e.g., HellaSwag_ML, ARC_ML) rely on this bridge for cross-lingual access, so masking TAHs leads to clear performance drops.
       - Translation-independent, reasoning-heavy tasks (e.g., XWinograd, XNLI) depend less on explicit alignment; in these settings, TAHs can introduce mild noise, explaining the non-monotonic pattern where pruning a moderate number of TAHs yields small regularization gains.
3. **Improved experimental rigor and analysis (Reviewers DWDG, ZKfU)**
   - **Systematic error analysis.** We added a qualitative error taxonomy (Appendix A.4) based on ablations of TAHs, identifying three dominant failure modes: No Translation (46%), Missing Details (36%), and Hallucination (18%). These patterns concretely illustrate the functional role of TAHs.
   - **Expanded model coverage.** To address concerns about model selection, we extended our experiments to four widely used multilingual model families (Llama 3, Gemma 2, Mistral, Qwen 3) spanning 1B–32B parameters, and included MoE architectures (Mixtral 8×7B, Qwen3-30B-A3B). Across all such settings, our main findings remain robust.
   - **Cross-lingual specialization (Reviewer ZKfU).** Following Reviewer ZKfU’s suggestion, we analyzed the specialization of TAHs across 20 language pairs. We found heads dedicated to “harder” language pairs and identified five TAHs used exclusively for en2X directions, which explains the higher overlap among en2X pairs compared with X2en pairs.

4. **Relationship to prior work (Reviewer DWDG)**

   We introduced a dedicated Related Work section (placed before the Conclusion) that more thoroughly situates our contribution within literature on multilingual capabilities and internal representations in LLMs, clarifying both connections and differences relative to prior studies.
5. **Figure quality and readability (Reviewers nLP1, DWDG, ZKfU)**

   We regenerated all figures as high-resolution vector graphics with larger fonts to improve readability.

During the rebuttal period, Reviewer DWDG noted that these additions resolved their main concerns.

We are grateful again for your thoughtful feedback and the opportunity to revise the paper, which we believe has significantly benefited from the review process.

Sincerely,

The Authors

---

### Meta-Review · Area_Chair_GhVY · 2026-01-19

**Summary:**

Reviewers generally found the core idea and analysis interesting, but also raised concerns about scope/terminology, mechanistic interpretation, and experimental rigor/positioning.

nLP1 (score 6) questions whether the term “translation heads” over-claims beyond token-level mapping;

DWDG (score 4) focuses on: representativeness of examples/model choices, figure readability, strength of data-augmentation gains, and completeness/placement of related work.

ZKfU (score 8) is positive but requests deeper analysis on language-pair specialization and clearer linkage to classic alignment/token mapping literature.

**Reviewer Concerns:**

During the rebuttal, it addressed:

nLP1: Over-claiming/terminology (“translation” vs alignment) addressed via renaming to TAHs; explanation for why alignment heads appear in middle layers (consistent with language-agnostic representations); clarification of mixed effects on other multilingual tasks.

DWDG: Concerns on rigor/presentation addressed by expanding evaluation across more models and sizes, adding more systematic analysis, improving related work, and regenerating figures; DWDG acknowledges the key issues are resolved.

ZKfU: Requests for language-pair specialization analysis and stronger linkage to alignment/token mapping literature addressed via expanded multi–language-pair analysis and updated related-work discussion.

Though there still remain concerns on the framing of the data-augmentation contribution raised by DWDG and the authors have agreed to soften the claims in wording.

**Reviewer Scores:**

DWDG explicitly indicates their main concerns are resolved and they would adjust the score more positively; remaining feedback is mainly wording/framing. So I think it will become all positive after the rebuttal and I recommend the acceptance.

---

### Decision · Program_Chairs · 2026-01-26

Accept (Poster)